# Structural roles of Ump1 and *β*-subunit propeptides in proteasome biogenesis

Eric Mark[1], Paula C Ramos[2], Fleur Kayser[1], Jörg Höckendorff[2], R Jürgen Dohmen[2], Petra Wendler[1]

**The yeast *pre1-1*(*β*4-S142F) mutant accumulates late 20S proteasome core particle precursor complexes (late-PCs). We report a 2.1 Å cryo-EM structure of this intermediate with full-length Ump1 trapped inside, and Pba1-Pba2 attached to the *α*-ring surfaces. The structure discloses intimate interactions of Ump1 with *β*2- and *β*5-propeptides, which together fill most of the antechambers between the *α*- and *β*-rings. The *β*5-propeptide is unprocessed and separates Ump1 from *β*6 and *β*7. The *β*2-propeptide is disconnected from the subunit by autocatalytic processing and localizes between Ump1 and *β*3. A comparison of different proteasome maturation states reveals that maturation goes along with global conformational changes in the rings, initiated by structuring of the proteolytic sites and their autocatalytic activation. In the *pre1-1* strain, *β*2 is activated first enabling processing of *β*1-, *β*6-, and *β*7-propeptides. Subsequent maturation of *β*5 and *β*1 precedes degradation of Ump1, tightening of the complex, and finally release of Pba1-Pba2.**

## Introduction

The 26S proteasome of eukaryotic cells serves essential functions in selective protein degradation, both for quality control and for regulatory purposes (Dikic, 2017). Moreover, the proteasome has emerged as a suitable drug target in the treatment of certain types of cancer and possibly other diseases (Fricker, 2020). The 26S proteasome is composed of a 20S catalytic core particle (CP), to the ends of which two 19S regulatory particles (RPs) are attached (Förster et al, 2013; Bard et al, 2018). The latter promote recognition of substrates, their unfolding, and translocation into the 20S CP. Alternatively, other activators such as Blm10/PA200 can bind to CPs to promote substrate processing (Schmidt et al, 2005; Iwanczyk et al, 2006; Bard et al, 2018). The CP represents the protease part of the 26S proteasome and is composed of two times 14 distinct subunits. It has a well-characterized cylindrical structure with a C2 symmetry (Groll et al, 1997). The cylindrical shape is formed by four stacked heptameric rings, with the two inner ones composed of *β*-subunits and two outer ones of *α*-subunits. The two *β*-rings surround a central chamber with six proteolytic active sites. The multicatalytic properties of this proteolytic machine are provided by the subunits *β*1, *β*2, and *β*5. Their N-terminal threonine (Thr[1]) residues serve as the nucleophiles that, together with conserved Asp[17] and Lys[33] residues, promote the peptide cleavage reaction. This mode characterizes the proteasome as an N-terminal nucleophile protease (Seemüller et al, 1996). All active CP subunits are initially synthesized in a precursor form with N-terminal propeptides (*β*1pro, *β*2pro, and *β*5pro) that are autocatalytically processed upon CP formation (Marques et al, 2009; Huber et al, 2016; Budenholzer et al, 2017). CP assembly is a very complex process involving multiple chaperones (Ramos & Dohmen, 2008; Murata et al, 2009; Budenholzer et al, 2017). It proceeds via the formation of half-proteasome intermediates lacking *β*7-subunits, known as 15S precursor complexes (15S-PCs) (Chen & Hochstrasser, 1996; Ramos et al, 1998). Incorporation of two precursor forms of *β*7 (*β*7pro) drives dimerization of two 15S-PCs and therefore formation of a 20S assembly precursor complex (Li et al, 2007; Marques et al, 2007). 15S-PCs consist of seven *α*-subunits, *β*3 and *β*4, propeptide-bearing precursor forms of the three catalytic *β*-subunits and of *β*6 (*β*6pro), and the chaperones Ump1 and Pba1-Pba2 (Li et al, 2007; Marques et al, 2007). In isolation, Ump1 is intrinsically unstructured (Sa-Moura et al, 2013; Uekusa et al, 2014). EM-based structural analyses complemented by cross-linking studies revealed that Ump1 is interacting with the inner surfaces of *α*- and *β*-subunits looping around the inner 15S-PC chamber (Kock et al, 2015; Schnell et al, 2021). The heterodimeric chaperone Pba1-Pba2, in contrast, is attached to the outside of the *α*-ring. An additional assembly chaperone, Pba3-Pba4, is involved in the formation of earlier intermediates but leaves the complex during 15S-PC assembly (Le Tallec et al, 2007; Yashiroda et al, 2008). Upon *β*7-driven dimerization of two 15S-PCs, Ump1 is encased in the nascent CP precursor complexes (Ramos et al, 1998; Kock et al, 2015; Walsh et al, 2023). What follows is a cascade of events, the exact order of which has remained unresolved, largely because they occur too fast to be followed in WT cells or complexes. These events include the autocatalytic processing of active *β*-subunits, the processing of *β*6pro

---

[1]Institute of Biochemistry and Biology, Department of Biochemistry, University of Potsdam, Potsdam-Golm, Germany   [2]Institute for Genetics, Center of Molecular Biosciences, Department of Biology, Faculty of Mathematics and Natural Sciences, University of Cologne, Cologne, Germany

Correspondence: j.dohmen@uni-koeln.de; pewendler@uni-potsdam.de
Jörg Höckendorff's present address is UCB BIOSCIENCES GmbH, Monheim, Germany

and β7pro, degradation of Ump1, and release of Pba1-Pba2 (Chen & Hochstrasser, 1996; Heinemeyer et al, 1997; Ramos et al, 1998; Groll et al, 1999; Kock et al, 2015; Wani et al, 2015). These maturation steps are impaired in yeast *pre1-1* cells carrying a mutant version of Pre1/ β4 (β4-S142F), which results in trapping of the encased Ump1 in this late CP precursor complex (Ramos et al, 1998; Kock et al, 2015), and impaired processing and activity of β-subunits (Heinemeyer et al, 1991, 1993). Rapid maturation of these precursor complexes precludes their isolation from wild-type cells.

By taking advantage of the *pre1-1* mutant, we now present a high-resolution cryo-EM structure of such a late precursor complex (late-PC) that reveals a near-complete structure of the two encased Ump1 molecules shedding light on their multiple interactions with the propeptides of β2 and β5 and with other proteasome subunits. Comparisons of the structures of different conformers of the late-PC with those of mature CPs provide insights into the order of events in the maturation cascade and the roles of Ump1 and propeptides in their control.

# Results

## A late 20S proteasome core particle assembly intermediate is trapped in the *pre1-1* mutant

We previously showed that proteasomal populations isolated from *pre1-1* cells yielded immature 20S CPs that either lacked the Pba1-Pba2 chaperone complex or were capped by it at only one of the two α-rings (Kock et al, 2015). We hypothesized that the mutation leads to slow maturation and therefore enrichment of proteasomal precursor complexes. This in turn causes a depletion of free Pba1-Pba2 chaperones, which cannot be properly recycled when the CP maturation process is impaired (Kock et al, 2015). According to this hypothesis, double-capped late-PCs represent an authentic assembly intermediate in the *pre1-1* mutant. In wt and *pre1-1* cells, the Pba1-Pba2 heterodimer is released upon maturation of 20S CPs and reused in further rounds of CP assembly (Kock et al, 2015; Wani et al, 2015). To address this issue in the *pre1-1* mutant, we overexpressed the genes encoding Pba1 and Pba2 (Fig S1A). As expected, this resulted in a useful yield of immature 20S core particles capped with Pba1-Pba2 at both ends. This allowed us to obtain high-resolution cryo-EM structures of this late core particle assembly intermediate, which we refer to as a late precursor complex (late-PC). Of all proteasomal complexes isolated via the FLAG-6xHis-tag on the β4-subunits from these cells, 41% were fully matured 20S CPs ([pre1-1]CP), 26% were late-PCs, 16% were 20S CPs with the Blm10 chaperone bound ([pre1-1]CP-Blm10), and 17% were 15S-PCs or other products (Fig S1B and C).

## Uncapped 20S CPs from *pre1-1* cells are fully matured

To understand how the *pre1-1*/β4-S142F mutation affects CP assembly, we solved the structure of the uncapped 20S [pre1-1]CP. The resulting [pre1-1]CP structure was resolved to 2.0 Å when C2 symmetry was applied during reconstruction (Figs 1A and S2A). The proteolytically active subunits in this structure are all autocatalytically cleaved, and the active sites are arranged similar to the structure of

the wt yeast CP (Fig 1B) (Huber et al, 2016). No chaperones or propeptide remnants are visible in the [pre1-1]CP structure. Our structure shows a wt arrangement of β5 active site residues (Thr[1], Asp[17], and Lys[33]) and autocatalytic proteolysis of β5pro consistent with the observation that the *pre1-1* strain has low chymotrypsin-like activity (Heinemeyer et al, 1991, 1993). We also find fully processed β5 in fractions containing mature proteasomes derived from the *pre1-1* strain in a gel filtration analysis (Fig S3A). Still, the chymotrypsin-like activity mediated by the β5-subunit is dramatically reduced in the *pre1-1* strain in comparison with wt. The trypsin-like and post-acidic activities, mediated by the β2- and β1-subunits, respectively, in contrast, are similar in wt and *pre1-1* cells (Fig S3B).

The overall structural effect of the β4-S142F mutation is minimal. Just like in the crystal structure of wt proteasomes, in the mature [pre1-1]CP, the β2 C-terminus wraps around β3 and is forming an antiparallel β-sheet with the β3-subunit (Fig 1C). The C-terminal residues of β7 insert between the β1'- and β2'-subunits of the opposite β-ring (Fig 1D), the C-terminus of β5 is contacting β3' and β4', and the β6 loop encompassing residues 155–175 is contacting the β2'- and β3'-subunit (Fig 1C).

The β4-S142F mutation is located at the interface between β4 and β5' of the opposite ring (Fig 1C and E). The phenylalanine is sandwiched between helix H3 in β4 and H3 in β5', which prevents helix H4, the S3/S4 β-hairpin loop (residues 18–34), and the H4/S9 loop (residues 172–177) in β4 from taking the wt positions in the *pre1-1* complex (Fig 1F). The S3/S4 β-hairpin loop and helices H3 and H4 of β-subunits are important for the assembly of the 20S complex in the eubacterium *Rhodococcus erythropolis*, where they are involved in assembly-mediated activation of the β-subunits (Witt et al, 2006). It is noteworthy that the active site residues Asp[17] and Lys[33] are located on the S3/S4 β-hairpin loop in β1, β2, and β5 of the eukaryotic proteasome. In the [pre1-1]CP structure, the S3/S4 β-hairpins of β3 and β5 assume the same position as seen in crystal structures of wt CP, whereas the hairpin of β4 is distorted and located near the central β-sheet of β5 (Fig 1F). We conclude that the *pre1-1*/β4-S142F mutation prevents β4 from reaching the correct fold in the mature [pre1-1]CP, despite being located at the β4/β5' interface and not near the disordered S3/S4 β-hairpin. The folding problems at the β4/β4' interface also result in an offset of the two 15S halves by ~2.5 Å (Fig S4, Video 1). A similar effect has been observed in a crystal structure of a catalytically inactive version (β-T1G) of the *Archaeoglobus fulgidus* CP where proteasomes fail to fully mature to a compact state because of the mutation (Groll et al, 2003). Aside from the described small differences, the overall structures of mature CPs from wt and *pre1-1* cells are very similar, with all active sites autocatalytically matured. This observation indicates that the mutation does not cause a general block of proteasome maturation, but rather reduces the efficiency of mature CP formation. The latter enabled the capturing and structural analysis of a late intermediate as described below.

## Ump1 and β-subunit propeptides are trapped in Pba1-Pba2-capped late core particle assembly intermediates

The increased abundance of late-PCs (doubly capped with Pba1-Pba2) allowed us to resolve the structure of this trapped late assembly intermediate to 2.3 Å without, and to 2.1 Å with C2 symmetry

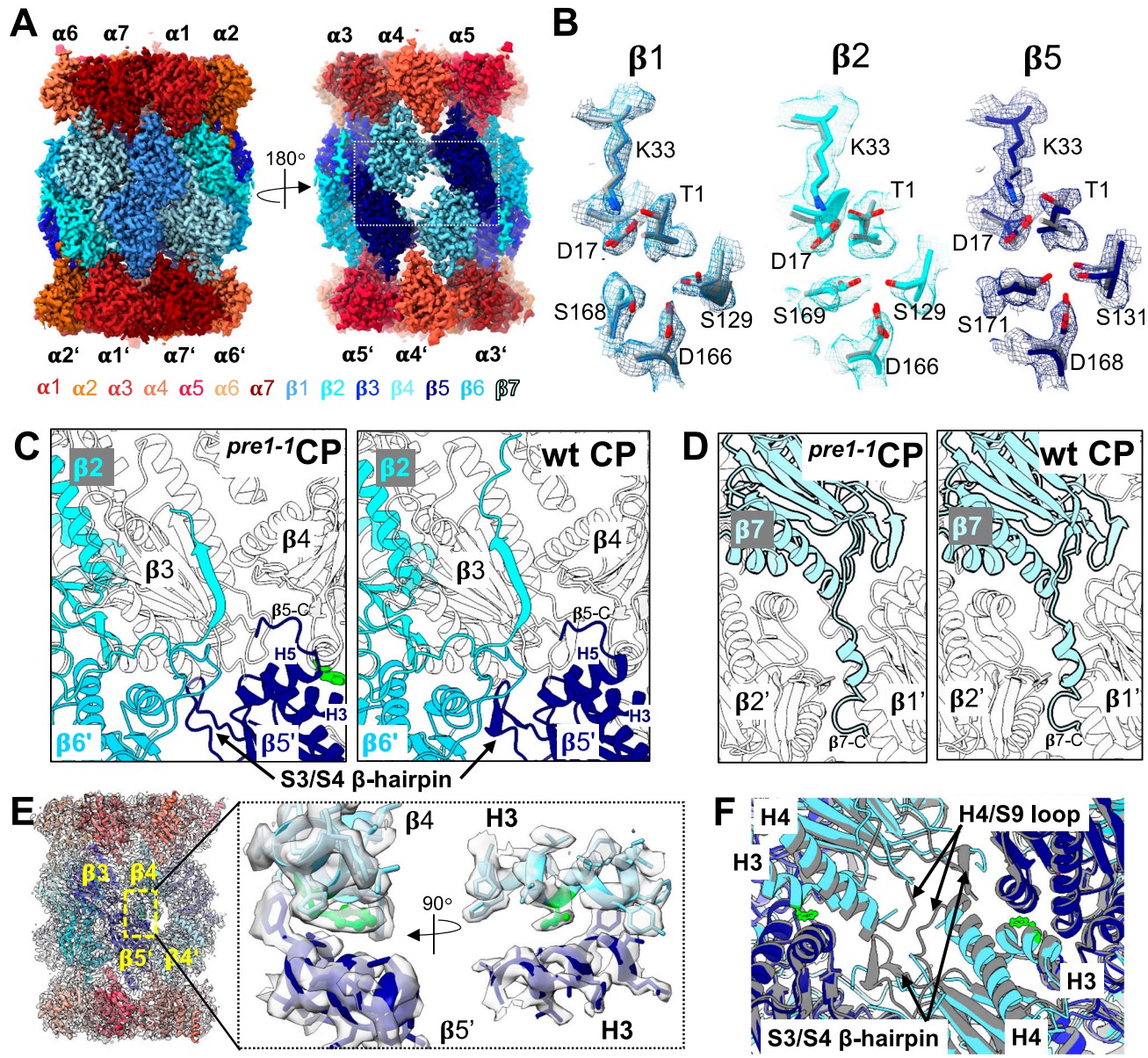

**Figure 1. Uncapped 20S CPs from *pre1-1* cells are fully matured.**
**(A)** 3D reconstruction of C2-symmetrized 20S CP from the *pre1-1* mutant (*pre1-1*CP) resolved to 2.0 Å shown as a side view (left) and frontal slab (right). **(B)** Superposition of the catalytic site residues from *pre1-1*CP (coloured) with wt CP (grey; PDB 5CZ4). Residues of *pre1-1*CP are also shown as a density map (mesh). **(C)** Interactions between the β2 and β5 C-termini in *pre1-1*CP (left) and wt CP structure (right; PDB 5CZ4). The S142F mutation is shown in green. **(D)** Interactions of β7-CTE with the β1′/β2′ interface in *pre1-1*CP (left) and wt CP structures (right; PDB 5CZ4). **(E)** Density map of *pre1-1*CP with a fitted PDB model at the site of the S142F mutation. **(F)** Conformation of the S3/S4 β-hairpins and the H4/S9 loops at the interface between β5-β4-β4′-β5′ in wt CP (grey; PDB 5CZ4) and in *pre1-1*CP (coloured) superimposed on the top ring β-subunit. F142 from *pre1-1*CP is shown as a green stick.

imposed during 3D reconstruction (Figs 2A, S1, and S2B). The differences between the maps obtained with and without symmetry imposed during reconstruction are marginal but will be addressed in a later section. The symmetrized map, however, was mainly used for data interpretation and confirmed the immature nature of the trapped assembly intermediate based on several criteria: (i) Pba1-Pba2 chaperone complexes are attached to the distal α-rings; (ii) two nearly completely resolved molecules of the Ump1 chaperone are encased in the complex; and (iii) β5pro and parts of β1pro and β2pro are observed (Fig 2A–C).

## Detection of full-length Ump1 reveals its many interactions in the late assembly intermediate

The nearly complete structures of the Ump1 chaperone and of β5pro, as well as parts of β1pro and β2pro, are resolved in the late-PC structure. As previously suggested by our cross-linking experiments (Kock et al, 2015), and in part consistent with a recent report (Walsh et al, 2023), the 148-residue Ump1 loops around the inside of the cavity formed between the α- and β-rings, contacting α1-4-, α7-, and all β-subunits except β6 (Fig 2B and C). In contrast to previously

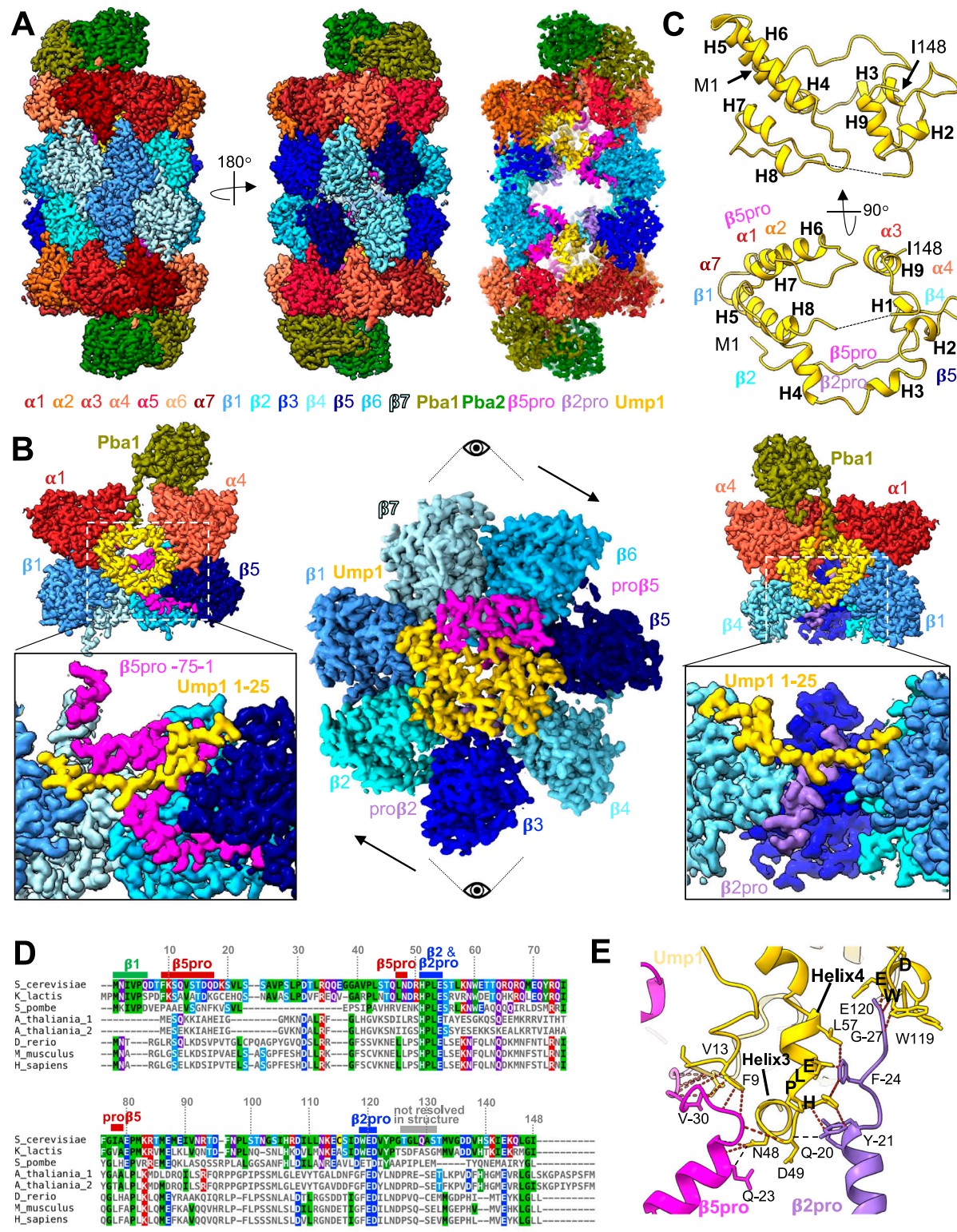

**Figure 2. Late-PCs reveal structures of full β5-propeptide and Ump1.**
**(A)** 3D reconstruction of C2-symmetrized late-PC from *pre1-1* mutant resolved to 2.1 Å shown as side views (left, middle) and central slab (right). **(B)** Top view of the isolated EM map densities of β-ring subunits and Ump1 of 3D reconstruction shown in A from the α-ring side (central panel). Side views onto the isolated map densities of indicated subunits highlighting interactions between the Ump1 N-terminus and β5pro (left), as well as β2pro (right). Points of view for left and right panels are given. **(C)** Structure of Ump1 with indications of interaction partners. Structural elements and termini are given. Residues 126–131 are indicated by a hyphenated line. **(D)** Alignment of Ump1 proteins from the indicated species. *A. thaliana* has two Ump1 paralogs. Residues of the *S. cerevisiae* Ump1 found to interact with β5pro or β2pro, or the mature parts of β1 or β2, as well as residues 126–131 (unresolved), are indicated. **(E)** Interaction network between β2pro, β5pro, and Ump1 sequences HPLE and WED.

reported precursor structures (Schnell et al, 2021; Walsh et al, 2023), the N-terminal 20 residues of Ump1 are well resolved in the late-PC structure. This segment lacks secondary structural elements and forms on one side hydrophobic contacts with helices H3-H5 (residues 21–66) of Ump1. Residues facing the cavity surface form hydrogen bonds with β1, β7, and β5pro (Fig S5A). The N-terminal six residues of Ump1 are in close contact with β1. An Ump1 residue Asp[7] forms a salt bridge with β7-Lys[106].

Between helices H8 and H9 of Ump1, residues 126–131 (Fig 2C) are not resolved in our late-PC structure. The six residues of this segment are not conserved between the Ump1 orthologues of *S. cerevisiae* and the related yeast species *Kluyveromyces lactis*. Nonetheless, the *K. lactis UMP1* gene, which encodes a protein with 61% identity to *S. cerevisiae* Ump1 (Fig 2D), can efficiently complement the severe growth defects of an *S. cerevisiae ump1Δ* mutant (Fig S6). These observations suggest that the sequence differences between KlUmp1 and ScUmp1, including residues 126–131, are not affecting structural features or interactions of the molecule critical for efficient proteasome assembly and maturation. The analysis of the late-PC thus revealed the structure of Ump1 encased in the antechamber of the structure where it engages in many interactions in particular with β2pro and β5pro.

### Extensive interactions between β5pro and Ump1

The structure of the late-PC reveals extensive contacts between parts of the Ump1 chaperone and the precursor form of the β5-subunit. Specifically, contacts are formed between residues –34 to –22 of β5pro and Ump1 residues 9–16 (Figs 2E and S5A). Despite a low sequence similarity between the β5pro precursors from yeast to humans (Marques et al, 2009), direct interactions between β5pro and its immunoproteasome counterpart β5ipro have previously been shown for the human orthologue hUmp1/POMP (Heink et al, 2005), suggesting that such interactions are functionally conserved. In addition, similar interactions have been reported recently between the hUmp1 residues 12–19 and the β5pro residues 37–40 (corresponding to –23 to –10 if one considers the first residue of the mature subunit as +1) in the structure of a half-CP precursor complex containing α1-α7, β1-β7 with PAC1-PAC2, and hUmp1 (PDB 8QYN) (Adolf et al, 2024). In contrast to previously reported precursor structures, all 75 residues of β5pro except –65, –64, and –13 are resolved in the late-PC structure (Adolf et al, 2024), β5pro reaches through the pore formed by the β-ring to the α7-subunit and is sandwiched in between Ump1, β6, and β7, presumably helping to coordinate β6 and β7 incorporation during the last steps of 15S-PC assembly (Fig 2B and C). The observation that β5pro is essential for viability of yeast cells in the presence of Ump1, but not in its absence, suggested that β5pro might be important to properly localize Ump1 in the complex (Ramos et al, 1998), and to promote the association of β6 and β7 with the Ump1-containing early precursor complexes. The fact that Ump1 establishes no contacts to β6 and only forms one salt bridge to β7 (Fig S4) substantiates the assumption that β5pro aids incorporation of β6 and β7 during the assembly process. Furthermore, β5pro and the N-terminal 20 residues of Ump1 have not been resolved in the structure of a precursor complex lacking β1 and β7 (Schnell et al, 2021). Incorporation of these subunits might hence be needed for β5pro and

the Ump1 N-terminus to adopt a conformation stable enough to be resolved by cryo-EM as is the case in the late-PC and the human half-CP precursor. The β5pro residues –63 to –28 interact with the helices H1 and H2 of β6 and β7 (Fig S5B). The functional relevance of these residues is supported by the observation that charged residues in this part of β5pro ([−49]ESD[−47] and [−7]KIK[−5]) have been characterized as particularly sensitive to mutations in systematic Ala scanning mutagenesis (Li et al, 2016). In summary, the β5pro separates Ump1 from β6 and β7 in the β-ring of the late-PC.

### Interaction of Ump1 with the precursor form of β2 involves conserved residues

Earlier work had suggested that β2 is the first β-subunit to assemble on the α-subunits during human CP assembly (Hirano et al, 2008). In this process, the incorporation of β2 depends on Ump1 and vice versa. In the late-PC structure, the β2-subunit has undergone autocatalytic processing leading to the exposure of the N-terminal active site Thr[1] residue. Nonetheless, a part of the cleaved-off β2pro (residues –27 to –12) is still resolved in the structure.

Strikingly, Ump1's contacts with β2 and its propeptide in the late-PC involve two strongly conserved amino acid stretches, [51]HPLE[54] and [119]WED[121] (Fig 2D and E). The β2pro is sandwiched between these conserved motifs, contacting the HPLE motif with Phe[−24] and Tyr[−21], and the WED motif with Ala[−28] and Gly[−27]. Hence, the N-terminus of β2pro is packed between Ump1 and β3 (Fig 2B and E). Sequence alignment of β2-subunits from various species shows that the few conserved residues (6 of 29) are either engaged in the aforementioned interactions with Ump1, in contact with β3, or directly preceding the active site Thr[1] residue (Fig S7). The apparent conservation of these β2-Ump1 interactions suggests that they might play a key role in the biogenesis of CPs in eukaryotes.

Overall, the structure of Ump1 overlays well with Ump1 residues 21–148, which were resolved in the Pba1-Pba2-bound 20S-PC from the *pre1-1 pre4-1* double mutant (Fig S8) (Walsh et al, 2023). However, two deviations from our structure are to be observed compared with the latter structure. Residues 39–42 and 46–49 of Ump1 deviate in their backbone position by up to 1.6 Å and 4.0 Å, respectively, from the structure obtained from the *pre1-1 pre4-1* mutant. Although Ump1-Ala[40] in our structure contacts β4-Tyr[98] and β5-Tyr[88], this residue is rotated away from the β4/β5 interface in the *pre1-1 pre4-1* 20S-PC structure indicating a slightly different binding of Ump1 to β4 and β5. The loop containing Ump1 residues 46–50 is positioned by interactions with β4-Ser[94], β5pro-Pro[−24], and β2pro-Gln[−20]. Degradation of β5pro, as is the case in the structure from the *pre1-1 pre4-1* mutant (Walsh et al, 2023), leads to the loss of hydrogen bonding of residues 46–48 and restructuring of the loop. This loop is located immediately before the conserved HPLE motif in Ump1 (Fig 2D and E), which positions β2pro with respect to β2 and β3. The Ump1 segment between residues 47 and 54 can thus convey information between the structural states of β2pro and β5pro within one 15S-PC half.

### β2 is the only catalytically active subunit in the late-PC

To deduce the maturation state of the late-PC, we analysed the active sites of the β-subunits (Fig 3A–C). Intriguingly, only the β2

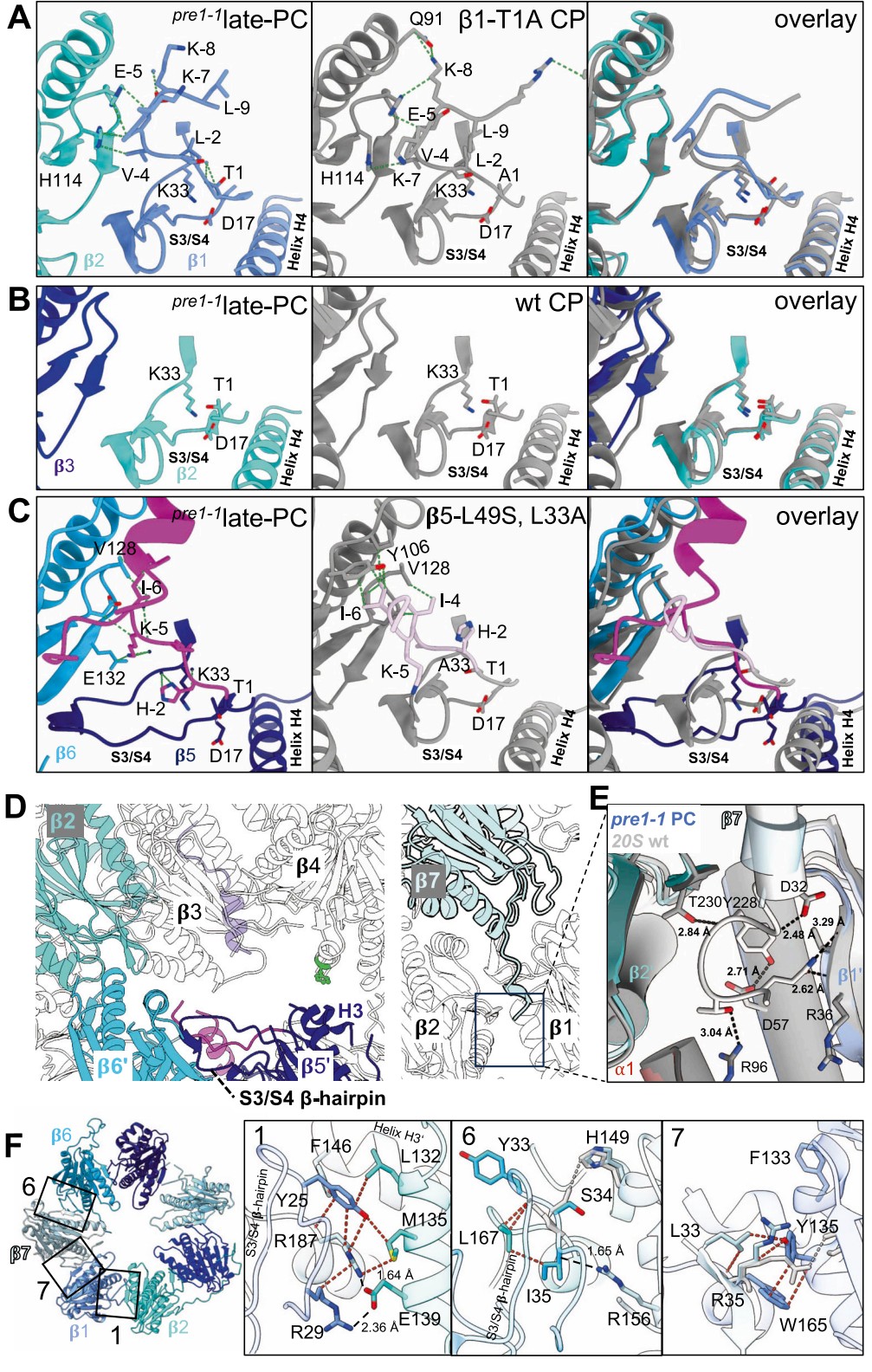

**Figure 3. Comparison of active site structures.**
**(A)** Structural superposition (right) of the subunit β1 of late-PC (left, coloured as in Fig 1) with 20S from a strain carrying a T1A mutation in β1 (middle; grey; PDB 5CZ5). Residues at positions that constitute the active site Asp[17], Lys[33], and Thr[1]/Ala[1], and precursor residues, are shown as sticks and coloured by heteroatoms. Contacts with van der Waals distances below 4 Å are indicated by green dashes. Only helix H4, the S3/S4 β-hairpin loop, and the propeptide are shown for β1. **(B)** Structural superposition (right) of the subunit β2 of late-PC (left) with wt 20S structure (middle; grey; PDB 5CZ4). Depiction of structural elements as in (A). **(C)** Structural superposition (right) of the subunit β5 of late-PC (left) with 20S from a strain carrying an L(-49)S and K33A mutation in β5 (middle; grey; PDB 5CZ8). Residues at positions that constitute the active site Asp[17], Lys[33]/Ala[33], and Thr[1], as well as precursor residues, are shown as in (A). The β5pro is shown in hotpink (late-PC) and lightpink (L(-49)S, K33A). **(D)** Interactions at the β-ring interface in the late-PC structure as seen for mature CP in Fig 1C (left) and Fig 1D (right). Only β2, β6′, β5′, and β7 are colour-coded as depicted in Fig 2. The S142F mutation is shown in green. **(E)** Interactions of β7-CTE with the β1/β2 interface. The viewing direction is as in (D). The late-PC (coloured, transparent) and the wt 20S structure (grey; PDB 5CZ4) are superimposed on β1. Residues involved in hydrogen bonds are depicted as sticks and coloured by heteroatoms. Hydrogen bond distances are given in Å. **(F)** Interactions at the β-ring interface of [pre1-1]CP between β1/β2/β7′ (1), β6/β7/β2′ (6), and β7/β1/β1′ (7) shown as indicated in the overview (left). The structure has been superimposed with the wt CP structure (PDB 5CZ4). Van der Waals distances are depicted in red dashed lines, and hydrogen bonds are given in black dashed lines, labelled by their distance in Å. Diverging wt side chains are shown in grey, and wt-exclusive interactions are depicted by dashed grey lines.

active site has a catalytically functional architecture after auto-catalytic cleavage of the Gly[-1]-Thr[1] peptide bond, which superimposes perfectly with the active site of the crystal structure of wt CP (Fig 3B). However, β2pro is still present in the structure with a segment between residues −1 and −11 missing. Consistent with these structural data, and much in contrast to β5, which remained

largely in the precursor form, a substantial proportion of β2 was found to be processed in fractions containing the late-PC after gel filtration (Fig S3A). β1pro is either flexible from the amino acid −9 or has been shortened by the adjacent β2 active site to 9 amino acids, which is consistent with data from active site mutants (Groll et al, 1999). The remaining propeptide is positioned similar to the one in a crystal structure of the CP carrying a β1-T1A mutation preventing autocatalytic activation of β1 (PBD 5CZ5; rmsd AS$^{-9 \text{ to } -1}$ = 2.9 Å) (Huber et al, 2016). As in the β1-T1A structure, the β1pro residues Glu$^{-5}$, Val$^{-4}$, and Leu$^{-2}$ occupy the S4, S3, and S1 pockets of β1 in the late-PC (Fig 3A). However, the side-chain orientations of Lys$^{-8}$, Lys$^{-7}$, Leu$^{-2}$, and Thr$^{1}$ subtly differ because β1 and β2 are shifted from the position in the mature CP, somewhat distorting the binding pockets. Of the proteolytically active subunits, the structure of the β5-subunit is most divergent from the mature state. The β5 residues Cys$^{-8}$, Ile$^{-6}$, Lys$^{-5}$, Ile$^{-4}$, and Ala$^{-3}$ occupy the β5-S6 and S4-S1 pockets, respectively (Fig 3C). This is reminiscent of the crystal structure of a β5-L(-49)S-K33A mutant, which contains β5pro remnants. However, these pockets are only rudimentary formed, because the C-terminal 87 residues of β5 and the S3/S4 β-hairpin loop are largely distorted in the late-PC (Fig S9). Helices H3 and H5 of β5, as well as residues 164–175, are disordered (Figs 1C and 3D).

Homologous regions in the 20S CP of R. erythropolis and A. fulgidus were found to be involved in eubacterial and archaeal proteasome maturation, respectively (Groll et al, 2003; Witt et al, 2006). Especially, the active site residue-bearing S3/S4 β-hairpin loop was suggested to act as an activation switch that couples the assembly of two 15S precursors with the formation of the proteolytic sites (Witt et al, 2006). A disorder of the S3/S4 β-hairpins in all β-subunits can also be observed in the 13S and pre-15S precursor complexes (Schnell et al, 2022). Similarly, our late-PC structure shows disorder of the S3/S4 β-hairpin loop not only in β5, but also in β3 and β4 (Fig S10). Parts of the loop are forming an antiparallel β-sheet with the right-hand neighbour in the β-ring, suggesting that the loop mediates a control mechanism to guide incorporation of the correct subunit during assembly. In accordance with the activation switch hypothesis (Witt et al, 2006), the active site of β5 in the precursor complex has not reached an active conformation yet, because the S3/S4 β-hairpin loop has not reached its final position (Fig 3C). In addition to the disorder in β5 of the late-PC, there is the disorder in the β2 C-terminus, the β6 loop encompassing residues 155–175, and the C-terminal residues of β7 (Fig 3D). The correct intercalation of the C-terminal extension (CTE) of the β7-subunit between subunits β1′ and β2′ in the opposing β-ring appears to activate β1, as shortening of β7 by 15 amino acids at the C-terminus renders β1-subunits in proteasomes of this strain inactive (Hilt et al, 1993; Ramos et al, 2004). In the late-PC structure, the last 10 residues of the β7-CTE are not resolved, whereas they are seen in the $^{pre1-1}$CP and the wt CP structures (Figs 1D and 3E). Given that the late-PC β1 active site residues resemble the active state, but autoactivation has not occurred yet, we conclude that residues 226–232 of β7′ in the opposite β-ring need to bind at the β1/β2 interface as seen in the wt CP and the $^{pre1-1}$CP to properly position β2, β1, and its propeptide for β1 activation.

The propeptides of β6 and β7 are flexible or have been cleaved to the mature state by β2, the only proteolytically active subunit in the late-PC structure. Because the 19 residues of β6pro and the 33

residues of β7pro are too far away from the β2-subunit in the same β-ring, both propeptides must be cleaved off by the β2′-subunit of the opposing β-ring. This has already been confirmed for wt cells by crystallographic studies (Groll et al, 1999). Remnants of a tryptophan- or tyrosine-containing octapeptide stemming from β2pro can be seen in proximity to the β2 active site. However, because these densities in the unsymmetrized late-PC map differed from those in the C2-symmetrized map, we analysed the sub-dataset using cryoSPARC's 3D variability analysis and obtained two distinct conformations we named late-PC1 and late-PC2 (Fig S1). The late-PC1 was resolved to 2.7 Å, whereas the late-PC2 was resolved to 2.3 Å without applying symmetry during 3D reconstruction (Table S1, Figs S1 and S2C and D). The most striking difference between the late-PC1 and late-PC2 is seen in map densities around the β2 active sites. Although the aforementioned octapeptide is located in both halves of the late-PC1 and late-PC2, another distinct density is found between β2pro and the octapeptide in both sides of the late-PC2 and just one side of the late-PC1 (Fig S11A and B). This density is modelled by ModelAngelo (Jamali et al, 2024) as PSGYT. Although the quality of the map in this area does not allow a clear assignment of amino acids, it is good enough to identify at least one bulky side chain (Tyr, Trp, and His). Because β1pro does not contain such amino acids (Fig S11C), we hypothesize that either β6′pro or β7′pro is the first substrate of β2 after autocatalytic activation. During 15S-PC dimerization, the incoming β6′- and β7′-propeptides would hence interact with β2pro in the opposite β-ring, promoting their maturation. In addition, the two 15S halves in the late-PC1 structure are further apart from each other than in the late-PC2 structure (Video 1). We conclude that late-PC1 is an earlier assembly state than late-PC2 and that the octapeptide represents β2pro remnants.

### Assembly checkpoints at the β-ring interface

The assembly of two half-proteasomes requires the correct interaction between β-subunits in opposing rings. For example, the growth defects caused by the pre1-1 mutation, which is located in the interface helix H3 of β4, can be alleviated by mutating C-terminal residues of the opposing β5-subunit (Chen & Hochstrasser, 1996). For the bacterial proteasome, Witt and colleagues have shown that hydrophobic interactions and salt bridges between β-subunit helices H3 and H4 at the interface of half-proteasomes drive proper positioning of the S3/S4 β-hairpin loop for autocatalytic activation of the proteolytic subunits (Witt et al, 2006). So far, the rearrangements during the assembly process could not be captured structurally. In the mature 20S CP, an intricate network of hydrophobic and electrostatic interactions forms at the interface, where two β-subunits of one ring interlock with one β-subunit of the opposing ring (Figs 3F and S10). For subunits β1 to β5, core interactions are similar and are explained below using β1 as an example. A hydrophobic residue at the tip of the S3/S4 β-hairpin loop, Tyr$^{25}$, interacts with hydrophobic residues on helix H3 of both the right-hand neighbour within the same ring (β2-Leu$^{132}$ and β2-Met$^{135}$) and the β-subunit (β7′-Phe$^{146}$) in the opposite ring. These hydrophobic interactions are extended to the aliphatic section of the side chain of a positively charged residue, Arg$^{187}$, from helix H4 in the subunit of the opposite ring. The charged section of this arginine residue interacts with an oxygen atom in the backbone of

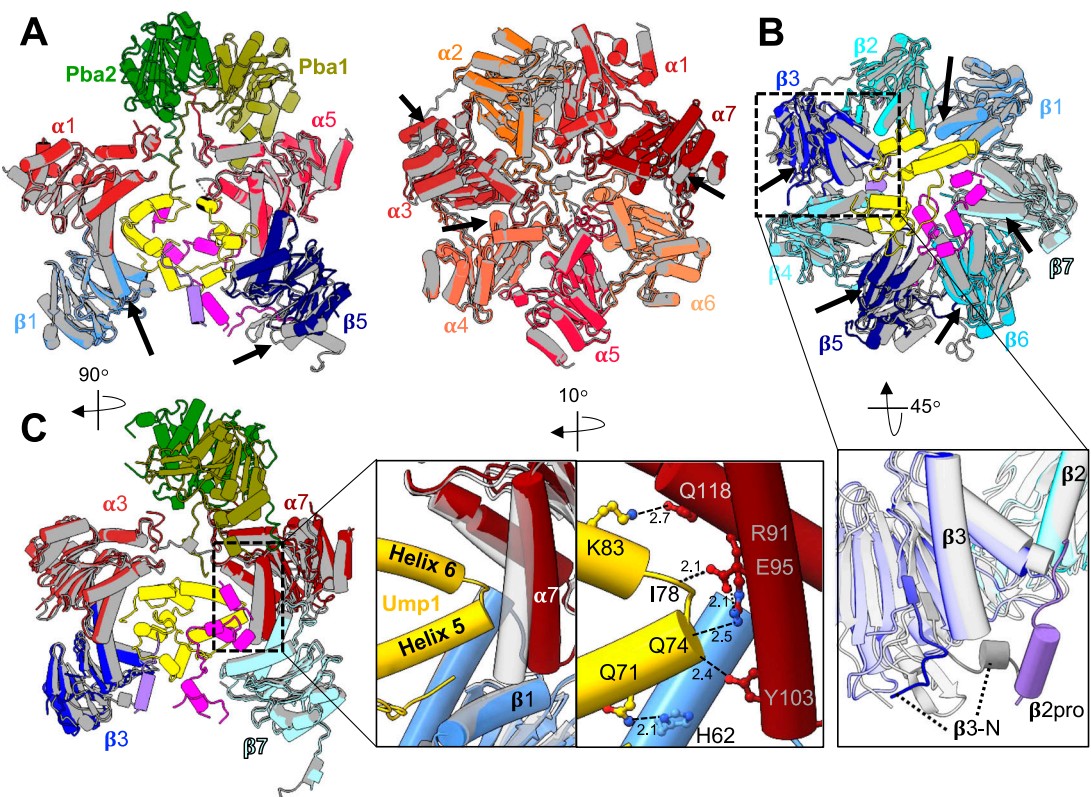

**Figure 4. Structural rearrangements in the late-PC structure.**
**(A)** Superposition of the late-PC structure (coloured as in Fig 1) onto the α-ring of the wt CP structure (grey; PDB 5CZ4) shown from the side displaying only Pba1-Pba2, Ump1, α1, α5, and β1, β5 for clarity (left), or showing only the α-ring (right) from the top. Arrows indicate areas of significant deviation between structures. **(B)** β-Ring structure in the same superposition as in (A) (top). Bottom, a close-up view of the β3-subunit. The displacement of the β3 N-terminus by β2pro is indicated. β4 is omitted for clarity. **(C)** Side view of superimposition shown in (A). Enlarged are contacts of Ump1 at the β1 and α7 interface. The wt structure is shown in transparent grey.

the S4 β-sheet of β1. In subunits β1 and β2, this interaction network is in addition supported by a salt bridge between charged residues on the S3/S4 β-hairpin loop and helix H3 of neighbouring subunits within the ring. The interaction network at the β6 and β7 S3/S4 β-hairpin loop differs from that of the other subunits insofar as the opposing subunits in the ring do not provide a positively charged residue that penetrates the plane of hydrophobic interactions. The interactions are predominantly of hydrophobic nature. Furthermore, van der Waals interactions are formed between β6-Ser[34] and β7-His[149] (Fig 3F). Arginine residues on both sides of β7 interact with backbone oxygen atoms in β-sheets of the left- and right-hand neighbour. This divergent interaction of β7 with neighbouring and opposing subunits could facilitate its incorporation as the last subunit into the ring. Apparently, this interaction is not strong enough to incorporate β7 into maturing 15S complexes, as β7 is unable to bind to complexes lacking β5pro (Li et al, 2007). Intriguingly, the comparison of the interactions at the S3/S4 β-hairpin loop interface of all β-subunits of the late-PC structure with the *pre1−1*CP mutant or wt CP shows that the β-hairpin loops of β1, β2, β6, and β7 have reached positions equivalent to the mature state (Fig S10). Albeit β3 has not assumed the final position relative to β2, autoactivation of β2 has already occurred as soon as the S3/S4 β-hairpin loop is structured into the mature state. This contrasts the situation in β1, which is not autocatalytically processed despite the correct positioning of all active site residues (Fig 3A and B). As

mentioned above, this might be due to regulation of β1 autocatalytic processing by the β7-CTE (Ramos et al, 2004).

In subunits β3-β5 of the late-PC, the central hydrophobic residues, which come to lie at the tip of the β-hairpin loop in the mature state, form an antiparallel β-sheet with the right-hand neighbour in the ring. Hence, without the central hydrophobic residue to assume position, loops and helices involved in interactions between both β-rings become disordered. Because of the β4-S142F mutation, this disordered state is maintained in β4 even in the mature state. These observations suggest that during assembly, the S3/S4 β-hairpin loops of β1-5 help the incorporation of the respective right neighbouring subunits into the ring by forming antiparallel β-sheets, which then need to restructure when the two 15S halves come together. Once this loop is restructured, the positively charged residues from the β-subunits in the opposite ring can assume their position, and all interfacing β-subunits can interlock.

Most importantly, restructuring will position residues Asp[17] and Lys[33] on the S3/S4 β-hairpin loop to assume their function in deprotonating the active site residue Thr[1] (Witt et al, 2006; Huber et al, 2016).

## Propeptides and Ump1 function together as a roadblock in maturation

In contrast to the eubacterial and archaeal proteasome, eukaryotic proteasomes need a chaperone, Ump1, for maturation. During

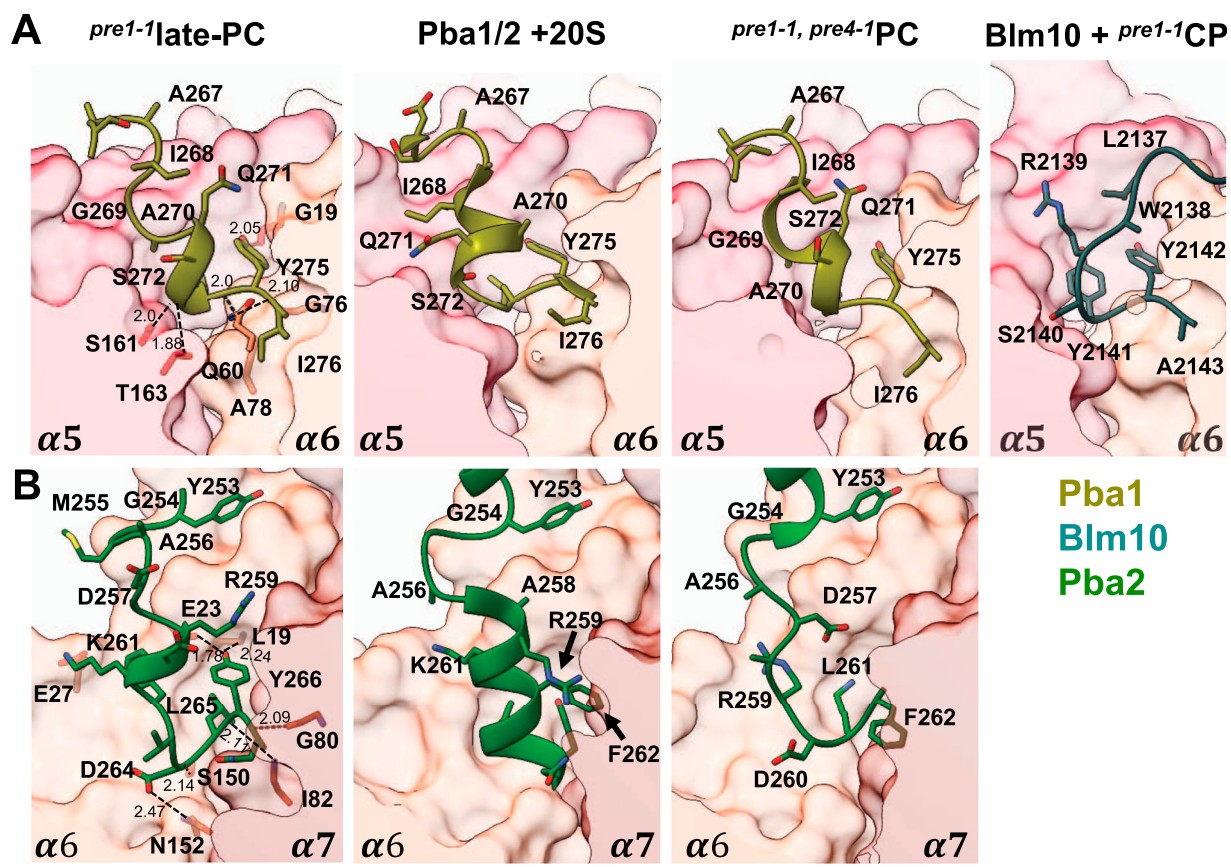

**Figure 5. Binding modes of HbYX motives of Pba1-Pba2 and Blm10.**
**(A)** Binding pocket of Pba1 or Blm10, respectively, at the α5/α6 interface in structures of the late-PC; reconstituted Pba1-Pba2-20S complex (PDB 4G4S); *pre1-1, pre4-1*PC (PDB 8T08); and Blm10-*pre1-1*CP (PDB 4V7O). All structures are superimposed on all α-ring subunits. **(A, B)** Binding pocket of Pba2 at the α6/α7 interface in the same structures as shown above in (A).

assembly of the 15S-PC, it binds to all of its subunits directly, except for α5, α6, and β6, which are bound by β5pro. Our late-PC structure indicates that Ump1, β2pro, and β5pro not only guide the ordered incorporation of subunits, but also function in expanding the PC to slow down the interlocking of α- and β-ring subunits. When structures of the late-PC of the *pre1-1* strain are superimposed on the α-ring subunits of the wt CP (Huber et al, 2016), subunits α3–α7, β1, β3, and β5–β7 do not align well (Fig 4A and B). In the α-ring of the late-PC, α3, α5, and α6 are shifted outwards, α4 is shifted towards α5, and α7 is shifted upwards towards Pba1-Pba2 by up to 5 Å. In the β-ring, β1 is shifted towards Ump1, β3 is shifted towards β4, and β5–β7 are shifted towards β4-β6. None of these shifts are seen in the mature *pre1-1*CP structure (Fig S12A); hence, the chaperones and propeptides must cause the displacement. Indeed, the N-terminus of the β3-subunit, which forms an α-helix in the wt CP, is displaced by β2pro in the late-PC and is reoriented towards the β3/β4 interface within the ring (Fig 4B). At the same time, all structural elements at the β-ring interface, including helix H3 of the β5-subunit in the opposite β-ring, are disordered in the late-PC (Figs 3D and S10). Thus, parts of β3 and β5 are shifted from their position in the wt state by up to 10 Å, leading to the separation of the β-rings at the β3/β6 interface by up to 8 Å (Fig S4, Video 1). Helices H5 and H6 of Ump1 are bound via multiple hydrophobic and charged

interactions at the interface between α7 and β1 (Fig 4C) preventing both subunits from taking their final positions. Overall, these structural deviations could contribute to the β7-CTE not being able to bind between β1 and β2 thereby impairing β1 autocatalytic processing.

Intriguingly, none of the shifts seen in α3, α7, β1, β3, and β7 of the Pba1-Pba2-bound late-PC can be observed in the reconstituted Pba1-Pba2-bound 20S complex (Stadtmueller et al, 2012) (Fig S12B), indicating that the insertion of Pba1 and Pba2 HbYX motifs into the α-ring pockets leads mainly to a local rearrangement of the neighbouring α-subunits to open the ring pore. In contrast to the crystal structure of the reconstituted Pba1-Pba2-20S CP complex (Stadtmueller et al, 2012) and to that of the Pba1-Pba2-bound precursor complex from the *pre1-1 pre4-1* mutant (Walsh et al, 2023), the Pba2 HbYX motif in the α6/α7 pocket of our structure adopts a position similar to the binding mode of Pba1 and Blm10 in the α5/α6 pocket (Fig 5A and B). Pba2-Tyr[266] is hydrogen-bonded with α6-Glu[23] and α6-Leu[19]. Four more hydrogen bonds are formed between the C-terminal four Pba2 residues and amino acids in α5 and α6 (Fig 5B). Furthermore, the interaction is stabilized by a salt bridge between Pba2-Lys[261] and α6-Glu[27], in accordance with the observation that Pba1-Pba2 binding is salt-sensitive (Stadtmueller et al, 2012). In comparison with the positioning of Pba1 or Blm10 in

the α5/α6 binding pocket, the HbYX motif of Pba2 in our map is shifted upwards in the binding pocket by 3 Å, presumably because neither α6 nor α7 has adopted their final position.

The binding mode of the Pba1 HbYX-containing loop to the α5/α6 pocket differs slightly between the reconstituted Pba1-Pba2-20S complex and all natively isolated precursor complexes (Fig 5A). In our late-PC structure, residues 267–273 of Pba1 are rearranged so that α5-Ser[161] and α5-Thr[163] establish hydrogen bonds with the Pba1 backbone. It was shown that Pba1-Pba2 has a higher affinity to immature CP as compared to mature CP (Wani et al, 2015).

This switch in affinity has been explained by the many interactions between the N-terminus of Pba1 when it traverses the central pore in the α-ring contacting Ump1 (with Pba1-Leu[2] and Pba1-Phe[3]) and all α-subunits via their extended N-termini (Walsh et al, 2023). However, the different binding modes of reconstituted Pba1-Pba2-20S in comparison with natively isolated precursors suggest that dissociation of Pba1-Pba2 is likely initiated by a loss of binding to Ump1 and to the binding pockets in the α-ring, both of which go along with full maturation of the 20S complex, rather than a loss of binding to the α-subunit N-termini. According to this hypothesis, the structural rearrangements in the HbYX motifs of Pba1 and Pba2 seen between the late-PC and the reconstituted Pba1-Pba2-CP complex reflect the state of maturation. They occur simultaneously with rearrangements and tightening of the α- and β-rings after maturation of the proteolytically active subunits, and degradation of all precursor peptides and Ump1. Therefore, maturation of the complex starts at the β-ring interface and proceeds by gradual degradation of the remaining precursor propeptides and Ump1, which block tightening of the α-ring. Once Ump1 is degraded, the α-subunits take their final positions, leading to a switch in the binding of the HbYX motifs and dissociation of Pba1-Pba2.

### Structure of Blm10-capped $^{pre1-1}$CP

The pool of proteasomal complexes isolated via the FLAG-6xHis-tag purification on the β4-subunit also contained $^{pre1-1}$CP-Blm10 complexes in sufficient amounts for 3D reconstruction. The copurified complex was resolved to 2.4 Å (Figs S1, S2E, and S13A) and represents the first structurally characterized native Blm10–proteasome complex isolated from yeast cells. Overall, the $^{pre1-1}$CP-Blm10 structure strongly resembles the crystal structure of the reconstituted 20S-Blm10 complex (Sadre-Bazzaz et al, 2010) (PBD 4V7O; rmsd 1.58 Å over half-proteasome + Blm10). In contrast to previous reconstituted structures, Blm10 caps only one end of the CP in the $^{pre1-1}$CP-Blm10 complex, reflecting a low abundance of Blm10 in vivo (Guan et al, 2020). Similar to the $^{pre1-1}$CP structure, the proteolytically active subunits in the $^{pre1-1}$CP-Blm10 are all autocatalytically cleaved and no propeptide remnants or Ump1 is encased inside the complex (Figs S2E and S13B). However, previous results from us and others suggest that Blm10 also associates with proteasomal precursor complexes (Fehlker et al, 2003; Lehmann et al, 2008; Kock et al, 2015), which might be too short-lived to be captured by our approach. The Blm10-bound α-ring in the $^{pre1-1}$CP-Blm10 structure is in an open-gate conformation similar to already-reported RP-bound CPs and the crystal structure of the reconstituted complex (Fig S13C) (Sadre-Bazzaz et al, 2010; Eisele et al, 2018). In comparison, the other α-ring in the $^{pre1-1}$CP-Blm10 structure, which is not capped with Blm10, resembles a closed-gate conformation. An overlay with the wt 20S CP structure shows that the α3-subunit is poorly resolved compared with the other subunits in the $^{pre1-1}$CP-Blm10 map, implying high flexibility in this area (Fig S13B and D). This high degree of flexibility was not observed for the α3-subunits in any of the late-PC structures or the $^{pre1-1}$CP structure. We conclude that this effect is mainly caused by the association of Blm10 with the $^{pre1-1}$CP. Furthermore, α4 and α5 are slightly shifted away from the ring centre. We hypothesize that Blm10 binding to one side of the mature CP primes the α-ring of the opposite side to bind activators.

## Discussion

The high-resolution cryo-EM structure of the late-PC, containing map densities for Ump1 and propeptide residues, reveals that both proteasomal antechambers are almost filled with polypeptides in the late stages of proteasomal biogenesis. Contrary to substrates, which are kept unfolded by the antechamber (Ruschak et al, 2010), the propeptides and Ump1 gain structure through interaction with the proteasome antechambers inside the wall.

β5pro wraps along β6, β7, and α7 on the inside of the chamber, and contacts Ump1 on each side of the cavity, close to β6 and α7 (Fig 2). This finding is in line with recently published structures of human Ump1 captured in a half-CP complex and a Pba1-Pba2-bound precursor from a β3-Δ205 mutant in yeast (Adolf et al, 2024; Velez et al, 2024). The close interactions of β5pro with β6 and β7 suggest a pivotal role of this propeptide in the recruitment of these last subunits to join during 15S precursor assembly (Fig 6A). Through contacts between the β5 precursor and Ump1, especially β5pro and the first 20 residues of Ump1, information on structural changes can be relayed to all subunits of the half-proteasome. Here, the extended doughnut-like structure of Ump1 enables contacts with β1, β2, β4, β5, and β7, and several α-subunits (Figs 2 and 6A). The interactions with the PC-associated chaperones introduce structural changes in the α-rings during assembly as we have already shown by cross-linking experiments (Kock et al, 2015). Pba1-Pba2 binding leads to an opening of the α-ring pore and restructuring of the α-subunit N-termini (Schnell et al, 2021, 2022). Ump1, on the other hand, binds in between α7 and β1, and presumably prevents α7 and β7 from taking their final positions in the Pba1-Pba2-bound late-PC. The propeptides and chaperones contribute to the shifts of many of the proteasomal subunits relative to their position found in the mature complex. At the same time, the subunits of the β-ring are restructured upon integration of the last subunit (β7) into the ring that goes along with the dimerization of two 15S-PCs (Fig 6B) (Li et al, 2007; Marques et al, 2007). The restructuring concerns structural elements at the interface between neighbouring and opposing β-subunits. Here, the S3/S4 β-hairpin loop has a pivotal role in sensing neighbouring subunits and initiating the assembly-mediated activation of the proteolytic subunits (Witt et al, 2006). As long as the restructuring event is not completed, we and others observed a separation of the 15S halves by some Ångstrom, indicating larger space requirements in the precursor complex (Groll et al, 2003). Interestingly, in our late-

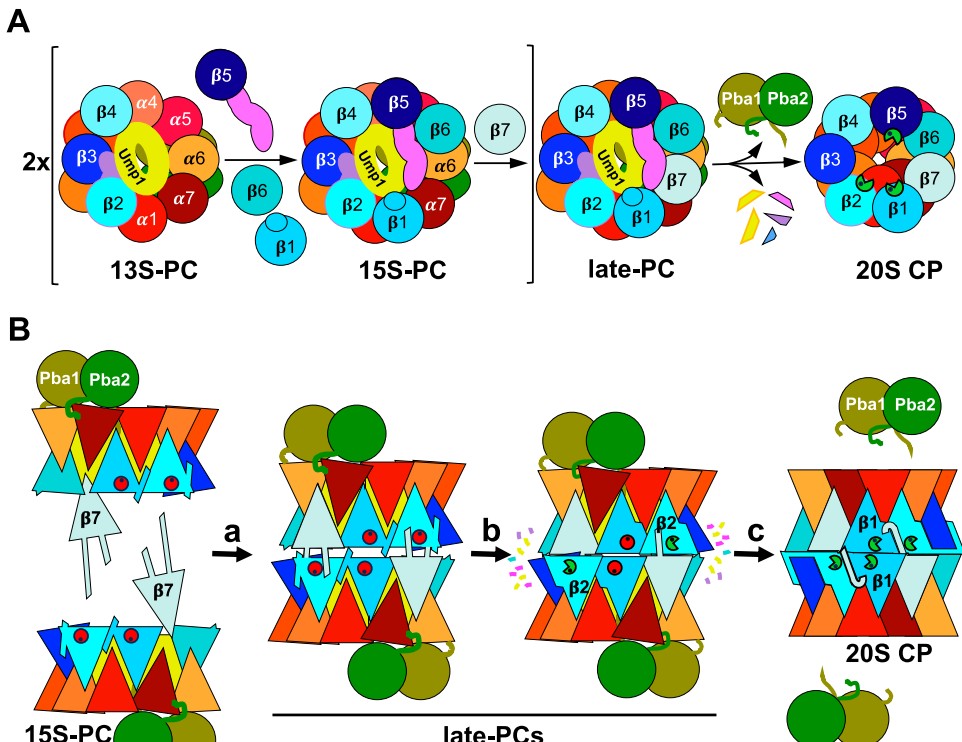

**Figure 6. Model of final stages of 20S proteasome assembly and maturation.**
**(A)** Structure-based schematic model emphasizing the roles of Ump1 and β-subunit propeptides in the assembly of 20S CP β-rings. The 13S-PC, a previously characterized precursor complex (Kock et al, 2015; Schnell et al, 2021), consists of a complete ring of α-subunits, the chaperones Pba1-Pba2 and Ump1, and subunits β2, β3, and β4. The 15S-PC intermediate is formed by the addition of β1, β5, and β6 (Kock et al, 2015; Schnell et al, 2021). The β5-propeptide has extensive interactions with Ump1 and β6, and promotes incorporation of the last β-subunit (β7), which drives dimerization of two 15S-PCs to form the late-PC. The latter complex is characterized by two Pba1-Pba2 chaperones attached to the α-rings, two encased Ump1 chaperones, and the presence of immature precursor forms of β-subunits. The propeptides of β2 (purple) and β5 (pink) frame the doughnut-shaped Ump1 located largely in the antechambers of the CP. Finally, processing of β-subunits results in formation of active sites leading to degradation of Ump1 and release of Pba1-Pba2, and thus to the formation of mature 20S CPs. Rearrangement of α- and β-rings going along with these maturation steps (for details, see (B)) is indicated in exaggerated form as a tightening of the rings. The cartoon depicts proteasome halves viewed from the β-ring interface. **(B)** Schematic representation of side views illustrating structural rearrangements during late steps in CP biogenesis. In step a, two 15S-PCs dimerize upon the addition of two β7 precursor subunits depicted with N-terminal propeptides and C-terminal extensions. In the resulting late-PC, β-subunits are initially inactive, the S3/S4 β-hairpin loops are oriented towards neighbouring subunits, and Ump1 is encapsulated inside (yellow). In step b, S3/S4 β-hairpin loops reorient. β2 and possibly β5 autoactivate and process propeptides of β1, β6, and β7. In step c, subunits of the two halves move into their final positions, close inter-ring interactions are established, β1 autoactivates, the β7-CTE locks into the β1/β2/α1 pocket, and enclosed Ump1 and propeptides are degraded leading to conformational changes causing release of Pba1-Pba2. Schematic representation of subunits in both parts of the figure follows the colour-coding used in the structures shown in Figs 1 and 2. Immature active sites are depicted as red circles, and mature active sites as green Pac-Men.

PC we observe successful restructuring in one half of the β-ring (β7/β1/β2), but not on the side where β3, β4, and β5 are located. The structural rearrangements involving the S3/S4 β-hairpin loop can therefore occur independently and are not concerted. It is known from the literature that proteasomal biogenesis is a step-by-step process that exhibits certain redundancies. For example, 15S-PC dimerization cannot proceed without incorporation of β7 into the complex, the Pba1-Pba2 chaperones have a certain retention time at the precursor complex, and autocatalytic activation of the proteolytic subunits requires the correct positioning of residues Lys[33] and Asp[17] in the active sites (Huber et al, 2016). However, the active subunits β1, β2, and β5 can be autocatalytically processed independently, at least to some extent without a fixed order, as suggested by structural and functional analyses of mutants wherein the natural order of events is impaired because of inactivation of individual active sites (Groll et al, 1999; Jager et al, 1999). Thus, there cannot be only one fixed pathway from precursor complexes to active proteasomes, but during maturation, it should be ensured that the proteolytically active subunits are not activated prematurely and that as few dead-end products as possible are formed. Therefore, the process must be accompanied by important checkpoints.

Based upon the comparison of the structures of late-PCs with mature CPs, we propose a model in which the presence of β-subunit

propeptides and Ump1 inside the PC blocks some subunits from taking their final positions until further maturation steps have occurred (Fig 6). Our late-PC1 and late-PC2 structures suggest that during β7-driven dimerization of two 15S half-proteasome precursor complexes, the maturation process initiates with autocatalytic activation of the β2-subunits. An unidentified density was observed in the complex with remnants of β2pro in just one half of the late-PC1 and both halves of the late-PC2. We speculate that this density might be remnants of β6′pro or β7′pro, which were cut by the active β2-subunit, as it was already suggested in previous studies (Groll et al, 1999). In addition, activation of β2 appears to go along with structural changes in the β-rings.

Based on this observation, we hypothesize that shortening of β1pro, β6pro, and β7pro might promote a closer association of both 15S-PC halves. Hence, activation of the proteolytically active β-subunits takes place before adjustments in the α-rings that allow for dissociation of the Pba1-Pba2 chaperone and association with the 19S RP. Even though β2 is the first subunit to be autocatalytically activated in the *pre1-1* mutant, β2 and β5 might be activated simultaneously in wt cells. Interestingly, however, we observed the impairment of β1 processing in a mutant with inactive β2 (β2-T1A), whereas a mutation causing inactivation of β1 had no detectable effect on β2 or β5 processing (Fig S14), which suggests an early role

of β2 activation in the activation cascade also in wt cells. This result is consistent with the earlier observation that upon fractionation of proteasomal complexes extracted from yeast cells, unprocessed β2pro is only detected in the 15S-PC and absent from later intermediates, whereas β1pro and β5pro are detected both in the 15S-PC and in larger complexes (Ramos et al, 1998). Notably, however, structural analyses of recombinant human proteasome assembly intermediates suggest that β1 and β5 might be processed before β2 (Adolf et al, 2024). Whether these apparent deviations reflect species-specific differences or are merely consequences of the different experimental set-ups remains unclear. The conserved segments of helices H3 and H4 in Ump1 act as a checkpoint and communicate the precursor states by restructuring upon the degradation of one of the propeptides as seen in the structure of the *pre1-1 pre4-1* mutant in comparison with our late-PC structure. The restructuring of Ump1 and the propeptides is also necessary for their degradation. This may be supported by the diverse interactions between these peptides, which enable them to pull each other along. The interfaces between both β-rings have to interlock properly for the C-terminal six residues of β7 to reach the β1/β2 binding pocket in the opposite ring. In contrast to β2 and β5, β1 is more dependent on the correct formation of the specificity pockets in the substrate binding channel, which seems to depend on this β7 interaction. Based on our model, autoactivation of β1 acts as a checkpoint and can only occur after repositioning of the S3/S4 β-hairpin loops in all subunits of the rings. The β-subunits and subsequently the α-subunits of the late-PC take their final positions in the complex upon gradual clearance of the chaperone and propeptides from the antechamber. The long Ump1 helices H5 and H6 are located in the late-PC directly next to β1 so that β1 is most likely responsible for their degradation. This would ensure that α7 only assumes its final position and thus ejects the Pba1-Pba2 chaperone when all β-subunits have taken their final positions and the proteolytically active subunits are matured. The S142F (*pre1-1*) mutation is located at the β4/β5′ interface and slows down 20S maturation by preventing β4 from forming a close interface with β4′, thereby hindering the restructuring of the β4 S3/S4 β-hairpin upon sensing the neighbouring subunits. It remains unclear why the β5 catalytic activity in the *pre1-1* mutant is severely reduced despite functional autolysis, because both reaction mechanisms involve the same residues (Huber et al, 2016). However, a similar observation is made for the *pre4-1* mutation in the *pre1-1* background, which also leaves the β1-subunit processed, but inactive (Walsh et al, 2023).

In contrast to what has been suggested based on a structural analysis of a mutant 15S-PC lacking β1 (Schnell et al, 2021), the structural data obtained from analysing the late-PC, which revealed the by and large full structures of Ump1 and β5pro, clearly show that there is remarkable agreement with early models of the interaction of these two polypeptides and its importance for proper proteasome biogenesis (Ramos et al, 1998). Our high-resolution structures specify these models by revealing that both the full-length Ump1 and large parts of β2pro and β5pro reside in the CP antechamber (Fig 6A). Consistent with this observation, trypsin treatment of 15S-PCs had already demonstrated that a large part of Ump1 is embedded within the structure formed by a complete α-ring and a β-ring lacking the β7-subunit (Ramos et al, 1998).

The finding that Ump1 resides nearly completely in the antechamber between the α- and β-rings, with only a small part exposed to the opening of the catalytic chamber formed by the two β-rings, raises the question as to how ultimately degradation of Ump1 is achieved. One possible mechanism might involve β5pro, in the absence of which, the presence of Ump1 causes a lethal block in proteasome biogenesis (Ramos et al, 1998). In this model, during proteasome maturation, β5pro would be important for bringing Ump1 into a position suitable for its degradation by the β-subunits. Further experiments that might employ versions of the complexes bearing, for example, mutant Ump1 or propeptides designed based upon the interactions observed in the late-PC structure will be required to test this idea.

# Materials and Methods

### Experimental model and study participant details

#### *Protein expression, purification, and analysis*
Proteasome populations were purified from the strain PR434, which is derived from the strain MO23 (Kock et al, 2015) and overexpresses Pba1-Pba2 under the GAL1 promoter. Protein complexes were prepared by a tandem affinity purification protocol using FLAG and 6xHis-tags as described before (Kock et al, 2015). Mass spectrometry was used to determine the composition of the separated bands by native PAGE. Analysis of the proteasomal chymotryptic (CT) activity was performed as follows: 90 μl of each fraction was mixed with 5 μg of the substrate succinyl-Leu-Leu-Val-Tyr-7-amido-4-methylcoumarin. Activities in 10 μg of crude extract proteins were determined as using the same reagent as mentioned above for CT activity, and Boc-Leu-Arg-Arg-AMC and Z-Val-Ala-Asp-AMC, respectively, for tryptic and post-acidic activities. The assay conditions were described previously (Ramos et al, 1998). For complementation studies, the yeast strain JD59 (*ump1Δ*) was transformed with *LEU2*-marked high-copy plasmids expressing UMP1 genes from various species under the control of P$_{CUP1}$. Constructs expressing *S. cerevisiae*, *S. pombe*, mouse, and human, as well as chimaeras composed of parts from *S. cerevisiae* and mouse *UMP1* genes, were described previously (Burri et al, 2000). *K. lactis UMP1* was cloned from a library of genomic *Eco*RI fragments after identification by colony hybridization using *S. cerevisiae UMP1* as a probe. Sequence analysis confirmed the presence of the sequence encoding an Ump1 orthologue. The *S. pombe UMP1* gene was amplified from cDNA and cloned into the same expression vector (pJDCEX2) as described previously (Burri et al, 2000).

Sequence alignments were created with Clustal Omega and illustrated with MView highlighting residues identical to *S. cerevisiae* Ump1. Sequence sources were as follows: gene locus SGD YBR173C/UMP1 (*S. cerevisiae*); NCBI XP_455055.1 (*K. lactis*); NCBI NP_587944.1 (*S. pombe*); NCBI NP_198681.1 and NP_564892.1 (*Archaeoglobus thaliania*); NCBI NP_001003424.1 (*D. rerio*); NCBI NP_079900.1 (*Mus musculus*); NCBI NP_057016.1 (*Homo sapiens*).

### EM data acquisition

5 μl of FLAG eluate (350 μg/ml) was applied to freshly glow-discharged Quantifoil R2/4 300-mesh holey carbon grids with

2-nm carbon support films. Glow-discharging of the grids was done in amylamine vapour. The protein solution was incubated on the grid for 45 s at 4°C and 85% humidity before blotting manually for 3 s and plunge freezing into liquid ethane. Cryo-EM images were collected on a Titan Krios transmission electron microscope (Thermo Fisher Scientific) operated at 300 kV equipped with a BioQuantum post-column energy filter (Gatan) and a K3 direct electron detector (Gatan). 29,904 micrographs were recorded in counting mode at a pixel size of 0.834 Å using SerialEM (Mastronarde, 2005). The defocus range was set between −0.6 and −2.0 $\mu m$. Each micrograph was dose-fractionated to 40 frames with a total exposure time of 2 s and a total electron dose of 44 e$^-$/Å$^2$.

### Cryo-EM image analysis

Image processing and 3D reconstruction were performed using cryoSPARC version 4.3 (Punjani et al, 2017). Movieframe alignment and dose weighting were performed with MotionCor2 (Zheng et al, 2017), and contrast transfer functions were determined using CTFFIND4 (Rohou & Grigorieff, 2015). All refinements used gold-standard Fourier shell correlation (FSC) calculations, and reported resolutions are based on the FSC = 0.143 criterion of mask-corrected FSC curves.

All maps were sharpened using automatically determined negative B-factors in cryoSPARC. If not otherwise stated, all processing steps were performed using cryoSPARC's standard settings. All described 3D reconstructions were performed using static masks. In addition, 3D reconstructions were performed by optimizing per-particle defocus and optimizing the CTF parameters per-exposure-group, namely, fitting beam tilt, beam tetrafoil, spherical aberration, beam tetrafoil, and beam anisotropic magnification. If not otherwise stated, all 3D classifications and 3D variability analysis were performed with particle stacks filtered to a resolution of 6 Å. All performed 3D refinement jobs were performed as masked refinements. The corresponding sterical masks were created according to the suggestions provided by cryoSPARC and using a dilation radius of 3 and a soft padding width of 10. The processing strategy is depicted in Fig S1. In short, low-quality micrographs with an estimated resolution over 5 Å, an estimated defocus over −3.0 $\mu m$, or a relative ice thickness over 2.0 were discarded. A total of 28,910 micrographs were used for image processing. 2D classification of 15,965 particles, picked via blob-picker, resulted in templates for template-based automated particle picking. A dataset of 10,556,408 particles was split into six equal-sized sub-datasets, which were subsequently subjected to several rounds of 2D classification to remove bad particles, resulting in a dataset of 1.389.861 high-quality particles of all proteasomal species. Four ab initio 3D models were generated of a particle subset containing 249,878 particles from the above-mentioned dataset. The dataset was split into two subsets, and heterogeneous refinement in cryoSPARC was used to separate the subsets into proteasomal species using the generated 3D initial models and limiting the resolution to 8 Å. The 3D classes obtained from both heterogeneous refinement jobs were merged, based on the proteasomal species they represent. The resulting proteasomal particle sets were further cleaned by removing duplicate particles and by a mixture of 3D classifications and 3D variability analysis in cryoSPARC (Punjani & Fleet, 2021) as described below. A total of 457,964 $^{pre1-1}$CPs were further cleaned by one round of 3D variability analysis. Non-uniform refinement in cryoSPARC of the final 341,154 20S $^{pre1-1}$CPs resulted in a reconstruction at 2.02 Å resolution when C2 symmetry was applied and 2.15 Å resolution when no symmetry was applied during 3D reconstruction (Fig S2A). A particle set of 394,211 particles, depicting the late-PC, was cleaned via one round of 3D classification with a resolution limit of 14 Å. The late-PC was reconstructed to a resolution of 2.14 Å using 233,748 particles and by applying C2 symmetry and 2.25 Å resolution when no symmetry was applied (Fig S2B). The same particle subset was further divided into two sub-populations using 3D variability analysis, named late-PC1 and late-PC2. The late-PC1 was reconstructed to a resolution of 2.69 Å using 53,919 particles without applying symmetry (Fig S2C). A particle subset containing 169,652 particles was used to reconstruct the late-PC2 at a resolution of 2.28 Å without applying any symmetry (Fig S2D). Finally, a particle set of 179,485 particles depicting the 20S $^{pre1-1}$CP-BLM10 was further cleaned via one round of 3D classification, one round of 2D classification, and one final round of 3D classification. The resulting 129,737 particles were used for a 3D reconstruction of the $^{pre1-1}$CP-Blm10 at a resolution of 2.39 Å without applying symmetry (Fig S2E).

### Molecular modelling

All PDB models were built using the implementation of ModelAngelo in Relion 5.0 (Jamali et al, 2024). Models were manually curated in COOT (Emsley et al, 2010) and refined using *phenix.real_space_refine* in Phenix (Liebschner et al, 2019). Water molecules were predicted using *phenix.douse*. Sequence predictions of unidentified map densities were done using *phenix.sequence_from_map* in Phenix. Structure visualization and comparison were done using UCSF ChimeraX (Goddard et al, 2018). The data collection and model statistics are summarized in Table S1.

### Data visualization

Figs 1A, B, and E and 2A and B were generated with maps that were post-processed using DeepEMhancer (Sanchez-Garcia et al, 2021). All hydrogen bonding distances are given as DH…A distances in Ångstrom unless stated otherwise. van der Waals interactions of atoms with centre–centre distances smaller than 4 Å are given as dashed lines without the distance label.

### Resource Availability

#### *Lead contact*
Further information and requests for resources and reagents should be directed and will be fulfilled by the lead contacts, Prof. Dr. Petra Wendler (pewendler@uni-potsdam.de) and Prof. Dr. Jürgen Dohmen (j.dohmen@uni-koeln.de).

### Material availability

This study did not generate new unique reagents.

## Data Availability

The EM maps of the late-PC, $^{pre1-1}$CP, $^{pre1-1}$CP-Blm10, late-PC1, and late-PC2 are deposited under accession codes EMD-19523, EMD-19529, EMD-51221, EMD-19527, and EMD-19528, respectively. Atomic coordinates and structure factors derived from the EM maps have been deposited in the Protein Data Bank under accession codes 8RVQ ($^{pre1-1}$CP), 8RVL (late-PC), 8RVO (late-PC1), 8RVP (late-PC2), and 9GBK ($^{pre1-1}$CP-Blm10). This study does not report the original code. The software used is listed below. Any additional information required to reanalyse the data reported in this study is available from the lead contact upon request.

## Supplementary Information

## Acknowledgements

This work was funded by the Deutsche Forschungsgemeinschaft (DFG, German Research Foundation) project numbers 406260942 (to P Wendler) and 442219341 (to P Wendler and RJ Dohmen). It was also supported by iNEXT-Discovery, project number 871037, funded by the Horizon 2020 programme of the European Union (to P Wendler). High-resolution EM data were collected at the Central European Institute of Technology (CEITEC). The authors would like to thank Dr. Jirka Novacek for data collection at CEITEC, Dr. Maximilian Voit for IT support, Kerstin Nürrenberg for technical assistance, Dr. Maria Nunes for strains, and Prof. Dieter Wolf for gifting the polyclonal anti-Pre2 antibody.

### Author Contributions

E Mark: data curation, software, validation, investigation, visualization, and writing—original draft.
PC Ramos: resources, validation, investigation, visualization, and writing—original draft.
F Kayser: resources.
J Höckendorff: resources and investigation.
RJ Dohmen: funding acquisition, investigation, writing—original draft and project administration.
P Wendler: data curation, supervision, funding acquisition, validation, investigation, visualization, project administration, and writing—original draft.

### Conflict of Interest Statement

The authors declare that they have no conflict of interest.

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
