## [Reviewer comments · Life Science Alliance]

Life Science Alliance

Structural roles of Ump1 and beta-subunit propeptides in proteasome biogenesis

Eric Mark, Paula Ramos, Fleur Kayser, Jörg Höckendorff, R. Jürgen Dohmen, and Petra Wendler

DOI: <https://doi.org/10.26508/lsa.202402865>

Corresponding author(s): Petra Wendler, University of Potsdam and R. Jürgen Dohmen, University of Köln

Review Timeline:

Submission Date:	2024-06-04
Editorial Decision:	2024-07-15
Revision Received:	2024-08-07
Editorial Decision:	2024-08-19
Revision Received:	2024-08-23
Accepted:	2024-08-23

Transaction Report:

July 15, 2024

Re: Life Science Alliance manuscript #LSA-2024-02865

Prof. Petra Wendler
University of Potsdam
Karl-Liebknecht Str. 24-25
Potsdam-Golm 14476
Germany

Dear Dr. Wendler,

Thank you for submitting your manuscript entitled "Structural roles of Ump1 and beta-subunit propeptides in late proteasome biogenesis" to Life Science Alliance. The manuscript was assessed by expert reviewers, whose comments are appended to this letter. We invite you to submit a revised manuscript addressing the Reviewer comments.

Thank you for this interesting contribution to Life Science Alliance. We are looking forward to receiving your revised manuscript.

Sincerely,

B. MANUSCRIPT ORGANIZATION AND FORMATTING:

Reviewer #1 (Comments to the Authors (Required)):

The article aims at better describing the last events in yeast 20S proteasome assembly, by taking advantage of a Pre1/ β 4 mutant (pre1-1) in which these late maturation steps are partially impaired and result in the accumulation of late assembly intermediates that cannot be quantitatively isolated from wild type cells. This accumulation can be further enhanced by overexpressing the assembly chaperones Pba1 and Pba2. The work is based on cryo-EM analyses of these intermediates and on a detailed comparison with known structures or assembly defects described in the literature for normal or assembly-impaired 20S proteasomes.

The results presented in the manuscript are based on extensive data processing and 3D modeling using specialized softwares. As I am not a structural biologist, I am not in a position to pertinently comment on data acquisition and structural interpretations. I apologize for agreeing too quickly to review this manuscript. I hope that my very general comment below can nevertheless be useful.

The main added-value of this work is to provide a clearer description of the position of the assembly chaperone Ump1 in the precursor complexes and of its interactions with various proteasome subunits, and particularly with β 2 and β 5 propeptides. This allows the authors to propose a model for the chronology of the late events of proteasome assembly, which implies a precise sequence of local and global conformational changes and defines a critical role for Ump1 and the β propeptides to control this sequence. For me, the arguments presented are compelling and the model represents an important insight into the mechanisms governing proper proteasome assembly.

However, since the analysis is based on the structure of a mutated proteasome precursor complex bound to overexpressed assembly chaperones, the manuscript does not fully address the question of whether the sequence is exactly the same in a wild type strain. Consequently, even if convincing, the model remains speculative, as illustrated by the fact that the authors have to make several assumptions to fill some gaps, and use in many cases the verb 'suggest' or the conditional tense to present their interpretations. To me, it would be extremely beneficial for the solidity of the conclusions if the predictions made by the model could be experimentally tested to validate it. It seems to me that the detailed map of the interactions of Ump1 and β propeptides with their neighbors may allow to design point mutations in a WT context that could be used to validate some of the proposed assembly checkpoints in the sequence of structural rearrangements occurring at the late steps of assembly. Such data would strongly strengthen the conclusions of the manuscript.

Nevertheless, the article represents an important contribution to our understanding of the last structural reorganisation events contributing to proper assembly of the 20S proteasome. If there is no objection on the proposed structural conclusions by a reviewer expert in structural biology, the new information presented in this manuscript deserves publication in my opinion.

Minor point: the article incidentally presents the structure of the Blm10-20S complex. I imagine, but it is unclear and should be precised, that the 20S complex is the pre1-1 mutant. In addition, in figure S1A, it is mentioned that the Blm10 containing band also contains Pba1, which is not seen in the structure. If it is suspected that Pba1 is a contamination due to comigration in native gel of another structure, this should be explained for clarity. In any case, it is unclear to me whether the accumulation of the Blm10-20S complex seen in Fig. S1 is due to the pre1-1 mutation or Pba1-Pba2 overexpression, or both. It would be useful to present in the figure not only the results of WT vs pre1-1 + Pba1-Pba2 overexpression, but also WT + Pba1-Pba2 overexpression and pre1-1 alone, and to briefly discuss why the complex Blm10-20S accumulates in the mutant.

Reviewer #2 (Comments to the Authors (Required)):

The mechanism of proteasome maturation has most recently been described in two NSMB papers (doi: 10.1038/s41594-024-01268-9 and doi: 10.1038/s41594-024-01262-1). In this manuscript, the authors report their use of a yeast mutation [pre1-1(β 4-S142F)] to capture four maturation intermediates including late PC1, late PC2, mature 20S-CP, and 20S-BLM10. The resolutions of the four cryo-EM maps are good, and the chaperone UMP1, β 5 propeptide, β 1-propeptides, and remnants of cleaved β 2-propeptide are clearly resolved in these maps. This work provides some new information and complements the recent NSMB reports. Listed below are a few concerns that should be addressed before publication.

- 1) The S3/S4 β -hairpin loop plays a pivotal role in sensing neighboring subunits and initiating the activation of the proteolytic subunits. This loop is only labeled in Figs. 1F and 3D. but they are not sufficient to show the function of this loop. Please consider making a better figure to demonstrate the function/mechanism of this loop.
- 2) Please show the structure around S142F and explain why this mutation blocks proteasome maturation process.
- 3) About the structure of Blm10-capped CP. In a similar published structure (PDB: 4V7O), Blm10 caps both ends of CP but only one end here. Is there any thought or explanation? Could proteasome maturation affect Blm10 binding? The authors (line 415) "hypothesize that Blm10 binding to one side of the mature CP primes the α -ring of the opposite side to bind activators or to loosen the pore as an exit for substrate peptides". Did the authors observe conformational changes at the opposing end?
- 4) Fig. 4 shows structural rearrangement in the late-PC structure compared to wild type CP. Consider compare this with the 15SCP? Is it possible the rearrangement is due to β 7 incorporation?
- 5) β 2-propeptide is the first one to be cleaved in this manuscript. But in human proteasome (NSMB, doi: 10.1038/s41594-024-01268-9), β 1-propeptide was removed earlier than β 2 (compare map5 and preholo 20S CP). this is a species difference, or the order of β 1, β 2 activation not critical?

Minor points

- 1) Line 65, "the exact order of which has remained unresolved, largely because they occur too fast to be followed" - Should be "too fast to"
- 2) Fig. S1, classification of CP, there are densities on top of α -ring in 3D classes with 81,252 and 35,558 particles. Are they from Pba1/Pba2 or Blm10?
- 3) Fig. S13 "ab" should be "a, b".
- 4) The paragraph "Electron microscopy data acquisition" in methods is repeated (lines 948 and 960).
- 5) Defocus range is different in Method and Table S1.
- 6) Fig. 2D, please label the residue number in alignment.
- 7) Fig. 5A - residues 267 to 273 of Pba1 described in line 391 should be labeled.
- 8) Hydrogens are shown in some figures with stick presentation but not in others. Please be consistent.
- 9) Lines 275, 757, 760 - What is "active side residues"? This term was used three times in the text.
- 10) Line 392: "so that α 5-Ser161 and -Thr163 establish hydrogen bonds with the Pba2 backbone". How do α 5-Ser161 and -Thr163 bind to Pba2? Is it a typo?
- 11) The validation reports are "not for manuscript review". The authors should submit the formal reports.
- 12) Please consider adding more details to the Fig. 6 illustration, perhaps by combining previously reported insights. This can be very useful to the readers.
- 13) The data statistics of 20S-BLM10 is not listed in Table S1.
- 14) It is hard to follow Fig. 3. Fig. 3A: H4 and S3-S4 described in the figure legend should be labeled. Fig. 3C: what is "active side residues" in the figure legend? It's not clear why the authors use 3 different structures to compare the 3 active sites. Fig. 3D: what is the color code? What is the box for? Where is H5 of β 5, as well as residues 164-175 described in line 256? It may be better to overlap two structures to prevent back-and forth tracking. Are β 1 and β 2 of the right panel in the same ring where β 7 resides? Very confusing. Fig. 3E: cannot tell which subunit(s) the labeled residues belong to. The color code is also confusing. Fig. 3F: cannot tell the labeled residues are from wild type or pre1-1CP.
- 15) Did the Ump1 and propeptide positions change between late-PC and 15SPC?

Response to reviewer comments

We thank the reviewers for their constructive critique on our manuscript, which prompted us to perform additional experiments and extensively revise our manuscript. We have revised figures 2D, 3E, 5, 6, S1, S2, S5, S8, S9, S10 and S13, included the new figure S14, and modified the manuscript text to address the reviewers' suggestions.

Below we specifically address each of the reviewers' comments point by point:

Reviewer #1

The article aims at better describing the last events in yeast 20S proteasome assembly, by taking advantage of a Pre1/ β 4 mutant (pre1-1) in which these late maturation steps are partially impaired and result in the accumulation of late assembly intermediates that cannot be quantitatively isolated from wild type cells. This accumulation can be further enhanced by overexpressing the assembly chaperones Pba1 and Pba2. The work is based on cryo-EM analyses of these intermediates and on a detailed comparison with known structures or assembly defects described in the literature for normal or assembly-impaired 20S proteasomes.

The results presented in the manuscript are based on extensive data processing and 3D modeling using specialized softwares. As I am not a structural biologist, I am not in a position to pertinently comment on data acquisition and structural interpretations. I apologize for agreeing too quickly to review this manuscript. I hope that my very general comment below can nevertheless be useful.

The main added-value of this work is to provide a clearer description of the position of the assembly chaperone Ump1 in the precursor complexes and of its interactions with various proteasome subunits, and particularly with β 2 and β 5 propeptides. This allows the authors to propose a model for the chronology of the late events of proteasome assembly, which implies a precise sequence of local and global conformational changes and defines a critical role for Ump1 and the β propeptides to control this sequence. For me, the arguments presented are compelling and the model represents an important insight into the mechanisms governing proper proteasome assembly.

However, since the analysis is based on the structure of a mutated proteasome precursor complex bound to overexpressed assembly chaperones, the manuscript does not fully address the question of whether the sequence is exactly the same in a wild type strain. Consequently, even if convincing, the model remains speculative, as illustrated by the fact that the authors have to make several assumptions to fill some gaps, and use in many cases the verb 'suggest' or the conditional tense to present their interpretations. To me, it would be extremely beneficial for the solidity of the conclusions if the predictions made by the model could be experimentally tested to validate it.

It seems to me that the detailed map of the interactions of Ump1 and β propeptides with their neighbors may allow to design point mutations in a WT context that could be used to validate some of the proposed assembly checkpoints in the sequence of

structural rearrangements occurring at the late steps of assembly. Such data would strongly strengthen the conclusions of the manuscript.

We are glad that the reviewer finds our arguments concerning the roles of Ump1 and propeptides in controlling the maturation processes of the 20S CP convincing and considers our model an important insight in the mechanism of proteasome assembly.

We generally agree with the reviewer's comment that one needs to keep in mind that the structures are obtained from a strain with a mutation in the *PRE1* gene, which is why we phrased our conclusions with the necessary caution, as noted by the reviewer. As we have addressed in some detail, however, because it is presently not possible to obtain such an assembly intermediate from a wild-type strain, because of its way too short half-life, we have to rely on other approaches such as the utilization of mutants as in our study or the generation of recombinant complexes as in the study on the human proteasome reported recently by Adolf et al. (2024) (DOI: <https://doi.org/10.1038/s41594-024-01268-9>). We believe that the results obtained with the *pre1-1* mutant version of the proteasome provides helpful indications of how the maturation process of the proteasome proceeds because it traps the chaperones Ump1 and Pba1-Pba2, as well as large parts of the propeptides without causing major structural distortions of the overall complex. We agree with the reviewer that structural knowledge of intermediate assembly/maturation states could be used to try to design point mutations and study their effects on the process. However, based upon our previous experience, we believe this to be a long shot with difficult to predict outcome, and as such beyond the scope of our current structural analysis. One reason is that results of introduced point mutations that are expected to block assembly steps are difficult to predict because multiple mutations e.g. in the propeptides are likely to be necessary to obtain detectable effects on proteasome assembly and phenotypes. On the other hand, the consequences of individual mutations are difficult to predict as they may affect not only the interaction seen in the structure, but also the tertiary structure of the individual polypeptide. In the past, we have used random mutagenesis of Ump1 to identify ones with detectable effects on proteasome function. Most of the identified mutants carried multiple mutations consistent with the above notion that single mutations effecting individual interactions might be insufficient to monitor detectable effects. One single mutation, however, affected residue W119 which was mutated to an arginine. This mutation caused a growth defect and a severe defect in $\beta 5$ maturation. W119 was seen to be in close contact with the propeptide of subunits $\beta 2$ (see figures 2D and 2E) consistent with the possibility that this interaction of Ump1 with the $\beta 2$ propeptide is important during proteasome assembly. While the strong effects of such mutations seem to be consistent with the observed conservation of affected residues and the detected interaction, they remain difficult to interpret and insufficient to unambiguously prove the contribution of observed discrete interactions in the structure for the assembly and maturation process. For this reason, we prefer not to elaborate on these still very preliminary findings in our current manuscript, and instead leave this open for future studies by us and others. We have added a statement mentioning the potential of such mutational studies as an outlook to the very end of the discussion section (line 550 ff).

Nevertheless, the article represents an important contribution to our understanding of the last structural reorganisation events contributing to proper assembly of the 20S proteasome. If there is no objection on the proposed structural conclusions by a

reviewer expert in structural biology, the new information presented in this manuscript deserves publication in my opinion.

We thank the reviewer for sharing our opinion that our findings provide a valuable contribution to our understanding of structural reorganisation taking place during 20S CP biogenesis.

Minor point: the article incidentally presents the structure of the Blm10-20S complex. I imagine, but it is unclear and should be precised, that the 20S complex is the pre1-1 mutant.

We thank the reviewer for this remark and adjusted our manuscript. The BLM10-capped complex is now referred to as ^{pre1-1}CP-BLM10.

In addition, in figure S1A, it is mentioned that the Blm10 containing band also contains Pba1, which is not seen in the structure. If it is suspected that Pba1 is a contamination due to comigration in native gel of another structure, this should be explained for clarity. In any case, it is unclear to me whether the accumulation of the Blm10-20S complex seen in Fig. S1 is due to the pre1-1 mutation or Pba1-Pba2 overexpression, or both.

As the reviewer suggests, the presence of Pba1 peptides detected by mass spectrometry in the same gel slice does not allow to conclude that Pba1 is present in the same complex as Blm10, even though this can also not be excluded. Such hybrid complexes, however, could not be identified in the cryo-EM analysis suggesting that, if they occur, they represent only a very minor fraction of the preparation.

It would be useful to present in the figure not only the results of WT vs pre1-1 + Pba1-Pba2 overexpression, but also WT + Pba1-Pba2 overexpression and pre1-1 alone, and to briefly discuss why the complex Blm10-20S accumulates in the mutant.

We have compared the abundance of Blm10-containing proteasomal complexes in yeast extracts of the different strains by native PAGE/anti-Blm10 western blot analysis (rebuttal figure 1). The data indicate that the abundance of doubly Blm10-capped complexes is increased in the pre1-1 background and not caused by overexpression of *PBA1* and *PBA2*. This finding is in line with earlier results indicating that Blm10 associates with proteasomal complexes resulting from a disturbed assembly (Lehmann et al. 2008; DOI: 10.1038/embor.2008.190). As expected and intended in our study, overexpression of Pba1-Pba2 efficiently reduced the amounts of 2xBlm10-capped complexes with a concomitant increase of free Blm10, and instead increased the amounts of Pba1-Pba2-capped complexes (as shown in Fig. S1A). As intended, the latter effect enabled purification of sufficient amounts of doubly-Pba1-Pba2-capped late-PCs for the cryo-EM analysis. Because we have not investigated this in sufficient detail, as the analysis of Blm10-containing complexes was only a observation at the side and not a main focus of the present study, we prefer not to include the results of these experiments in the manuscript. Based upon these and earlier findings (Kock et al. 2015; DOI: <https://doi.org/10.1038/ncomms7123>), however, we can conclude that Blm10-containing proteasomal complexes generally are detected in higher amounts when assembly is disturbed and Pba1-Pba2 becomes limiting, which can therefore be reduced by overexpression of *PBA1* and *PBA2* genes.

Reviewer #2 (Comments to the Authors (Required)):

The mechanism of proteasome maturation has most recently been described in two NSMB papers (doi: 10.1038/s41594-024-01268-9 and doi: 10.1038/s41594-024-01262-1). In this manuscript, the authors report their use of a yeast mutation [pre1-1(β 4-S142F)] to capture four maturation intermediates including late PC1, late PC2, mature 20S-CP, and 20S-BLM10. The resolutions of the four cryo-EM maps are good, and the chaperone UMP1, β 5 propeptide, β 1-propeptides, and remnants of cleaved β 2-propeptide are clearly resolved in these maps. This work provides some new information and complements the recent NSMB reports. Listed below are a few concerns that should be addressed before publication.

We thank the reviewer for the constructive criticism of our work and are glad that she/he attests to the good resolution of our maps, which include Ump1 and parts of β -subunit propeptides, concluding that our work as such complements recently published work.

1) The S3/S4 β -hairpin loop plays a pivotal role in sensing neighboring subunits and initiating the activation of the proteolytic subunits. This loop is only labeled in Figs. 1F and 3D. but they are not sufficient to show the function of this loop. Please consider making a better figure to demonstrate the function/mechanism of this loop. We agree with the reviewer that the S3/S4 β -hairpin loop is a crucial feature in proteasome biogenesis and we show this loop in many figures. As the reviewer pointed out correctly, the loop has not always been labeled. We have now labeled the loop in Figures 1C, 1F, 3A-D, 3F, S9, and S10. For clarity and to visualise the different functions of this loop we have also extended Figure S9 to show the interactions with neighbouring subunits.

2) Please show the structure around S142F and explain why this mutation blocks proteasome maturation process.

We have shown the S142F mutation and its structural effects in Figures 1E, 1F, 2D, S4, S10 and Movie 1 of the original manuscript. In the text, line 113-139, we also explain our observations of the effects of the mutation. The β 5 propeptide in the β 4 S142F mutated strain is still autocatalytically processed and the subunit shows residual β 5 activity, hence the mutation does not block maturation completely, but slows it down.

We conclude this in line 129ff:

“We conclude that the pre1-1/ β 4-S142F mutation prevents β 4 from reaching the correct fold in the mature pre1-1CP, despite being located at the β 4- β 5' interface and not near the disordered S3/S4 β -hairpin.

The folding problems at the β 4- β 4' interface also result in an offset of the two 15S halves by approximately 2.5 Å (Figure S4, Movie 1).”

and in line 137 ff:

“This observation indicates that the mutation does not cause a general block of proteasome maturation, but rather reduces the efficiency of mature CP formation.

The latter enabled the capturing and structural analysis of a late intermediate.”

Considering all of this, we hope that the reviewer will agree that we have addressed the structural and functional implications of the S142F mutation in sufficient detail.

3) About the structure of Blm10-capped CP. In a similar published structure (PBD:

4V7O), *Blm10 caps both ends of CP but only one end here. Is there any thought or explanation?*

We noticed these differences as well and explain them with the different nature of our approaches. The group that published the 4V7O structure used an *in vitro* approach incubating purified 20S CPs with purified BLM10 (Iwanczyk et al. 2006; DOI: <https://doi.org/10.1016%2Fj.jmb.2006.08.010>). A similar approach was used by another research group that published a structure of a human 20S CP-PA200 complex. This group elaborated on the importance of the molar ratio between PA200 and 20S CPs (Guan et al. 2020; <https://doi.org/10.1371%2Fjournal.pbio.3000654>). They noticed an increase of 20S CPs doubly capped with PA200 by increasing the molar ratio of PA200 to human 20S CPs from 4.4 to 8.8. This implies the importance of a larger surplus of PA200 or BLM10 in order to trigger the formation of doubly capped complexes. In comparison, we co-purified BLM10-capped complexes directly from the *pre1-1* containing yeast strain overexpressing Pba1-Pba2. As discussed above in response to a comment by the other reviewer, we rarely detect doubly Blm10-capped CPs in this strain. Doubly Blm10-capped CPs, by contrast, were detected in *pre1-1* extracts when Pba1-Pba2 was not overexpressed (see rebuttal figure 1). Since our focus was on the characterization of Pba1-Pba2-capped late PCs, we have not used the latter strain for purification of complexes in our current study. Furthermore, our native PAGE analysis indicated that singly Blm10-capped CPs are the predominant species of such complexes in wild-type cells. We now mention in the manuscript that the *pre1-1*CP-BLM10 complex was co-purified with the other proteasomal complexes and that it constitutes the first structure of a native Blm10-bound proteasome complex isolated directly from yeast cells (line 419 ff).

Could proteasome maturation affect Blm10 binding? The authors (line 415) "hypothesize that Blm10 binding to one side of the mature CP primes the α -ring of the opposite side to bind activators or to loosen the pore as an exit for substrate peptides". Did the authors observe conformational changes at the opposing end?

We and others (Fehlker et al. 2003, DOI: <https://doi.org/10.1038/sj.embor.embor938>; Marques et al. 2007, DOI: <https://doi.org/10.1074/jbc.M705836200>; Lehmann et al. 2008, DOI: [10.1038/emboj.2008.190](https://doi.org/10.1038/emboj.2008.190); Weberruss et al. 2013, DOI: <https://doi.org/10.1038/emboj.2013.192>) have found that Blm10 binds to alpha rings of mature as well as immature 20S particles. Our current hypothesis is that maturation defects lead to a higher proportion of accessible alpha rings (in particular if Pba1-Pba2 is scarce), which in turn will lead to more Blm10 bound complexes. We have not found any Blm10-bound precursor complexes in our cryo-EM analysis. The Blm10-bound half of the *pre1-1*CP-BLM10 structure in our manuscript very much resembles the published crystal structures, but the $\alpha 3'$ subunit of the opposite half displays very weak density indicating an area of high flexibility. This effect was not observed in the other maps presented here, which suggests that it is triggered by Blm10 binding. In the revised manuscript, we now explain our findings in more detail (line 431 ff).

4) Fig. 4 shows structural rearrangement in the late-PC structure compared to wild type CP. Consider compare this with the 15SCP? Is it possible the rearrangement is due to $\beta 7$ incorporation?

We agree with the reviewer that the comparison with the 15S structure would be very interesting in order to pinpoint what effect the incorporation of $\beta 7$ has on the complex. Unfortunately, we don't have an isotropic 15S structure in our dataset. The

group of John Hanna has published a pre15S complex (lacking the $\beta 1$ and $\beta 7$ subunits) from a *S. cerevisiae* strain containing a pre3-1 mutation (Walsh et al. 2023, DOI: <https://doi.org/10.1038/s41594-023-01081-w>), but this study did not publish an associated mature 20S structure. However, if we overlay the structures #4 (pre15S, no $\beta 1$ and $\beta 7$) and #5 (half 20S, with $\beta 1$ and $\beta 7$) of the human proteasomal precursor complex with the respective mature 20S complex (Adolf et al. 2024; DOI: <https://doi.org/10.1038/s41594-024-01268-9>), we observe similar re-arrangements in both precursor structures (rebuttal figure 2), suggesting that the widening of the alpha ring is indeed caused by the presence of Ump1 and the propeptides and not triggered by $\beta 7$ incorporation.

5) $\beta 2$ -propeptide is the first one to be cleaved in this manuscript. But in human proteasome (NSMB, doi: 10.1038/s41594-024-01268-9), $\beta 1$ -propeptide was removed earlier than $\beta 2$ (compare map5 and preholo 20S CP). this is a species difference, or the order of $\beta 1$, $\beta 2$ activation not critical?

This point by the reviewer is well taken. Our structural results indeed indicate that the $\beta 2$ subunit is the first to be activated in yeast. Our data show that active $\beta 2$ may contribute to the activation of $\beta 1$ by shortening its propeptide, which is consistent with earlier reports using active site mutants that revealed that active $\beta 2$ can shorten the propeptides of $\beta 1$ and $\beta 7$ (Groll et al. 1999; DOI: 10.1073/pnas.96.20.10976; Heinemeyer et al. 1997; DOI: 10.1074/jbc.272.40.25200; Huber et al. 2016, DOI: <https://doi.org/10.1038/ncomms10900>) and an earlier proposed two-step model of maturation of some beta subunits (Groettrup et al. EMBO J. 15, 6887-98, 1996). Seeking further evidence to support this conclusion regarding an early role of $\beta 2$, we have analysed the maturation of the three active subunits $\beta 1$, $\beta 2$, and $\beta 5$ in $\beta 1$ -T1A and $\beta 2$ -T1A mutants. As is now shown in the new figure S14, we found, consistent with the above-mentioned notion, that the $\beta 2$ -T1A mutation caused an impairment of $\beta 1$ maturation, whereas no converse effect was observed in the β -T1A mutant. Thus, several lines of evidence support our conclusion that $\beta 2$ is the first subunit to be processed in yeast. The most recent data obtained for the human proteasome, by contrast, suggest that $\beta 1$ pro and $\beta 5$ pro are cleaved off before $\beta 2$ pro in the maturation state captured in the preholo proteasome structure. Considering the highly dissimilar propeptides of yeast and human beta subunits, it is very well possible that these differences reflect species-specific differences.

However, we also want to point out that the structures of human CP intermediates were obtained by recombinant expression in insect cells and purification taking advantage of C-terminal twin-Strep tags on either $\beta 2$ or $\beta 7$, whereas our native yeast complexes were purified from *pre-1-1* yeast cells using a FLAG-6His tag on $\beta 4$. Notably, none of the proteasomal subunits in the structures of the human pre-holo proteasome contained any additional mutation that could result in a slowed maturation process. The last steps of proteasomal maturation are known to occur too fast to be captured with complexes extracted from wild-type cells. The possibility to obtain a sufficient amount of the human preholo 20S PC with the approach based upon recombinant expression in insect cells could therefore be an indication that maturation of this intermediate might be slower than in human cells for unknown reasons with possible impacts on the maturation processes. In any case, we agree that it is worth to note these apparent differences observed with the two experimental systems, which is why we pointed them out in the revised discussion (line 507 to 510)

Minor points

1) Line 65, "the exact order of which has remained unresolved, largely because they occur too fast to be followed" - Should be "too fast to"

Thank you. We have corrected this error.

2) Fig. S1, classification of CP, there are densities on top of α -ring in 3D classes with 81,252 and 35,558 particles. Are they from Pba1/Pba2 or Blm10?

These 3D classes specifically should contain a mixture of pre^{1-1} CPs, pre^{1-1} CPs with one Pba1-2 dimer attached to one of the α -rings and also some 15S PCs (and other PCs). Further processing of the particles in these classes resulted in small particle stacks with poor quality.

3) Fig. S13 "ab" should be "a, b".

Thank you for spotting this mistake. We fixed this typo.

4) The paragraph "Electron microscopy data acquisition" in methods is repeated (lines 948 and 960).

Thank you for pointing this out. We have deleted one of the sections.

5) Defocus range is different in Method and Table S1.

Thank you, we have fixed this error.

6) Fig. 2D, please label the residue number in alignment.

We have labeled the alignment with residue numbers in the revised Figure 2.

7) Fig. 5A - residues 267 to 273 of Pba1 described in line 391 should be labeled.

We have labeled all relevant Pba1 residues in Fig 5A of the revised manuscript now.

8) Hydrogens are shown in some figures with stick presentation but not in others. Please be consistent.

We thank the reviewer for spotting this inconsistency. We have now unified the stick representation to not show any hydrogens, changing Fig4C, Fig5, Fig S5, and Fig S8.

9) Lines 275, 757, 760 - What is "active side residues"? This term was used three times in the text.

We refer to Thr1, Asp17 and Lys 33 when mentioning active site residues. This has been stated in the figure legend of Figure 3, but we now clarify at the first mentioning, which residues we refer to (line 106).

10) Line 392: "so that α 5-Ser161 and -Thr163 establish hydrogen bonds with the Pba2 backbone". How do α 5-Ser161 and -Thr163 bind to Pba2? Is it a typo?

We thank the reviewer for this remark. We fixed this typo.

11) The validation reports are "not for manuscript review". The authors should submit the formal reports.

We have now uploaded the final reports for all structures submitted to the EMDB and PDB. This also includes the ^{pre1-1}CP-BLM10 structure.

12) Please consider adding more details to the Fig. 6 illustration, perhaps by combining previously reported insights. This can be very useful to the readers. In our model cartoon, we had tried to focus on the late steps of the assembly pathway to illustrate the conclusions derived from the structural analysis by us and others, but we welcome the suggestion to add further steps for which recent structural studies have already provided important contributions. To this end, in the revised version of figure 6A, we now include a representation of an earlier assembly intermediate, the 13S PC, which is converted to the 15S PC by stepwise addition of subunits $\beta 5$, $\beta 6$, and $\beta 1$. We have extensively revised the figure legend, which now includes references to previous studies on the earlier intermediates.

13) The data statistics of 20S-BLM10 is not listed in Table S1. We have now included the statistic for ^{pre1-1}CP-BLM10 into Table S1.

14) It is hard to follow Fig. 3. Fig. 3A: H4 and S3-S4 described in the figure legend should be labeled. Fig. 3C: what is "active side residues" in the figure legend? It's not clear why the authors use 3 different structures to compare the 3 active sites. Fig. 3D: what is the color code? What is the box for? Where is H5 of $\beta 5$, as well as residues 164-175 described in line 256? It may be better to overlap two structures to prevent back-and forth tracking. Are $\beta 1$ and $\beta 2$ of the right panel in the same ring where $\beta 7$ resides? Very confusing. Fig. 3E: cannot tell which subunit(s) the labeled residues belong to. The color code is also confusing. Fig. 3F: cannot tell the labeled residues are from wild type or ^{pre1-1}CP. We thank the reviewer for his/her constructive feedback on Figure 3.

We now have labeled helix H4 and the S3-S4 β - hairpin in Fig 3A-C, adjusted the figure legend to explain which residues are shown, fixed the box and labels of $\beta 2'$ and $\beta 1'$ in Figure 3D, explained the color code in the figure legend of Figure 3D, changed the color code of Figure 3E, and replaced Figures F1/6/7 with versions that show the color code of the subunits. We furthermore explain the reasoning for using different structures to compare the active sites in lines 247 ff. and lines 254 ff. of the revised manuscript. In brief, we used yeast derived X-ray structures for comparison that also show precursors of the respective active sites.

15) Did the Ump1 and propeptide positions change between late-PC and 15SPC? We do not observe significant differences in the positions of the resolved areas in Ump1 and the propeptides (rebuttal figure 3). However, in contrast to the available 15S PC structures, our late-PC structure has a fully resolved Ump1 N-terminus and a nearly fully resolved $\beta 5$ propeptide. In addition, the last 11 residues of the $\beta 2$ propeptide are only resolved on one side of our late-PC1 structure. This propeptide aligns well with the $\beta 2$ propeptide of the 15S PC.

Rebuttal figure 1: Western Blot analysis of yeast extracts from indicated strains separated by a 4-16% Bis-Tris Native Gel (Invitrogen) PAGE. Blm10 was detected using a polyclonal rabbit anti-Blm10 antibody (kind gift of Dr. Cordula Enenkel). Free Blm10, single Blm10 capped and doubly Blm10 capped CP are indicated.

Rebuttal figure 2: Overlay of the $pre1-1CP$ structure with human precursor structures published in Adolf et al. 2024. Overlay of structure #4 (coloured) with mature 20S structure (grey) in similar views as depicted in Figure 4 C (A) and Figure 4B (B) of the submitted manuscript. Overlay of structure #5 with mature 20S structure in similar views as above (C,D).

Rebuttal figure 3: Overlay yeast precursor structures. Overlay of the *pre3-1* pre15S structure (light grey, Schnell et al. 2021; pdb 7LS6) with the *pre1-1*CP (dark grey) and late-PC (coloured) structures presented in the submitted manuscript. The structures are depicted in similar views as in Figure 1A.

August 19, 2024

RE: Life Science Alliance Manuscript #LSA-2024-02865R

Prof. Petra Wendler
University of Potsdam
Karl-Liebknecht Str. 24-25
Potsdam-Golm 14476
Germany

Dear Dr. Wendler,

Thank you for submitting your revised manuscript entitled "Structural roles of Ump1 and beta-subunit propeptides in proteasome biogenesis". We would be happy to publish your paper in Life Science Alliance pending final revisions necessary to meet our formatting guidelines.

- please be sure that the authorship listing and order is correct
- Please upload all figure files individually, including the supplementary ones; all figure legends should only appear in the main manuscript file
- please add a Running Title to our system
- please add the Twitter handle of your host institute/organization as well as your own or/and one of the authors in our system
- please consult our manuscript preparation guidelines <https://www.life-science-alliance.org/manuscript-prep> and make sure your manuscript sections are in the correct order
- please upload a clean manuscript file without the track changes
- please upload your Table in editable .doc or Excel format
- we encourage you to revise the figure legends for Figure S5 such that the figure panels are introduced in alphabetical order
- there is a call out for figure S4A on pg. 6, and this figure has only one panel -- please correct
- please add callouts for Figures S1A-C and S11A-B to your main manuscript text;

LSA now encourages authors to provide a 30-60 second video where the study is briefly explained. We will use these videos on social media to promote the published paper and the presenting author (for examples, see <https://docs.google.com/document/d/1-UWCfbE4pGcDdcgzcmiuJl2XMBJnxKYeqRvLLrLS08s/edit?usp=sharing>). Corresponding or first-authors are welcome to submit the video. Please submit only one video per manuscript. The video can be emailed to contact@life-science-alliance.org

A. FINAL FILES:

B. MANUSCRIPT ORGANIZATION AND FORMATTING:

Sincerely,

Reviewer #2 (Comments to the Authors (Required)):

The authors have satisfactorily addressed our comments and concerns. This is a very nice piece of work.

August 23, 2024

RE: Life Science Alliance Manuscript #LSA-2024-02865RR

Prof. Petra Wendler
University of Potsdam
Karl-Liebknecht Str. 24-25
Potsdam-Golm 14476
Germany

Dear Dr. Wendler,

Thank you for submitting your Research Article entitled "Structural roles of Ump1 and beta-subunit propeptides in proteasome biogenesis". It is a pleasure to let you know that your manuscript is now accepted for publication in Life Science Alliance. Congratulations on this interesting work.

DISTRIBUTION OF MATERIALS:

Again, congratulations on a very nice paper. I hope you found the review process to be constructive and are pleased with how the manuscript was handled editorially. We look forward to future exciting submissions from your lab.

Sincerely,
